# ToolACE: Winning the Points of LLM Function Calling

Weiwen Liu[†1], Xu Huang[†3], Xingshan Zeng[†2], Xinlong Hao[2], Shuai Yu[2], Dexun Li[2], Shuai Wang[2], Weinan Gan[2], Zhengying Liu[2], Yuanqing Yu[5], Zezhong Wang[6], Yuxian Wang[4], Wu Ning[4], Yutai Hou[4], Bin Wang[2], Chuhan Wu[*2], Xinzhi Wang[2], Yong Liu[2], Yasheng Wang[*2], Duyu Tang[4], Dandan Tu[4], Lifeng Shang[2], Xin Jiang[2], Ruiming Tang[*2], Defu Lian[*3], Qun Liu[2], and Enhong Chen[3]

[1]Shanghai Jiao Tong University
[2]Huawei Noah's Ark Lab
[3]University of Science and Technology of China
[4]Huawei Technologies Co., Ltd
[5]Tsinghua University
[6]The Chinese University of Hong Kong
wwliu@sjtu.edu.cn, zeng.xingshan@huawei.com, xuhuangcs@mail.ustc.edu.cn

## Abstract

Function calling significantly extends the application boundary of large language models (LLMs), where high-quality and diverse training data is critical for unlocking this capability. However, collecting and annotating real function-calling data is challenging, while synthetic data from existing pipelines often lack coverage and accuracy. In this paper, we present ToolACE, an automatic agentic pipeline designed to generate accurate, complex, and diverse tool-learning data, specifically tailored to the capabilities of LLMs. ToolACE leverages a novel self-evolution synthesis process to curate a comprehensive API pool of 26,507 diverse APIs. Dialogs are further generated through the interplay among multiple agents, under the guidance of a complexity evaluator. To ensure data accuracy, we implement a dual-layer verification system combining rule-based and model-based checks. We demonstrate that models trained on our synthesized data—even with only 8B parameters—achieve state-of-the-art performance, comparable to the latest GPT-4 models. Our model and a subset of the data are publicly available at https://huggingface.co/Team-ACE.

## 1 Introduction

Equipping Large Language Models (LLMs) with external tools has significantly enhanced the capability of AI Agents to solve complex real-world tasks Huang et al. (2024); Qin et al. (2023); Qu et al. (2024). The integration of function calling enables LLMs to access up-to-date information, perform delicate computations, and utilize third-party services, thereby unlocking a wide range of potential applications across various fields, *e.g.*, workflow automation Zhong et al. (2023), financial reporting Theuma & Shareghi (2024), and travel planning Hao et al. (2024).

Function calls in real-world applications are often diverse and complex, driven by the varied functionalities of APIs[1] and the broad range of tasks Qin et al. (2023). APIs often undergo rapid updates to meet diverse user needs, necessitating models capable of robust zero-shot generalization. Additionally, users' requirements can be complex or ambiguous, leading to scenarios where multiple tools are employed in a parallel or dependent manner, or require multi-turn interactions. This highlights the importance of managing intricate instructions and accommodating various function-calling scenarios.

Despite these challenges, current tool-augmented LLMs primarily focus on simple function-calling tasks with limited diversity and complexity Qu et al. (2024). They mainly rely on existing public APIs for task construction, which restricts their zero-shot capabilities and applicability to single-turn queries, neglecting more complex scenarios such as dependent or multi-turn interactions Qin et al.

---

[†]Equal Contributions. [*]Corresponding authors.
[1]In this paper, APIs, tools, functions, and plugins are used interchangeably.

Table 1: Comparison of ToolACE with other representative tool-augmented LLMs (n/a represents not available.). ToolACE comprehensively incorporates the broadest range of APIs and domains, supports complex nested parameters (Nested), accommodates both parallel (Parallel) and dependent (Dependent) function calls, and addresses various types of tool-related data (Multi-type).

| Model | #API | #Domain | Nested | Parallel | Dependent | Multi-type |
|---|---|---|---|---|---|---|
| Gorilla Patil et al. (2023) | 1645 | 3 | ✗ | ✗ | ✗ | ✗ |
| ToolAlpaca Tang et al. (2023) | 3938 | 50 | ✗ | ✗ | ✗ | ✗ |
| ToolLLM Qin et al. (2023) | 16464 | 49 | ✗ | ✗ | ✓ | ✗ |
| Functionary Meetkai (2024) | n/a | n/a | ✗ | ✓ | ✗ | ✗ |
| xLAM Liu et al. (2024) | 3673 | 21 | ✗ | ✓ | ✗ | ✗ |
| Granite Abdelaziz et al. (2024) | n/a | n/a | ✗ | ✓ | ✗ | ✓ |
| **ToolACE** | **26507** | **390** | ✓ | ✓ | ✓ | ✓ |

(2023); Tang et al. (2023); Liu et al. (2024). Table 1 provides an overview of the data statistics used in these representative tool-augmented LLMs. Moreover, executions of function calls demand precise API selection and parameter configuration, which are highly dependent on the quality and accuracy of underlying data. As data becomes increasingly diverse and complex, generating accurate samples with simple pipelines introduced by the existing work becomes significantly more challenging.

In this paper, we present ToolACE, a systematic tool-learning pipeline that automatically synthesizes *accurate*, *diverse*, and *complex* function calling data, with the awareness of the model's capability.

**Evolutionary Diversity.** Exposing LLMs to a broad range of function-calling scenarios enhances their proficiency and zero-shot tool usage Zhang et al. (2024). Instead of relying on public APIs, ToolACE introduces a Tool Self-Evolution Synthesis (TSS) method. TSS uses a speciation-adaptation-evolution process to generate tools across multiple domains with diverse data types and constraints. Starting with pretraining data to ensure comprehensive coverage, this iterative process of self-evolution and continual updates expands the diversity of the API pool, enabling more sophisticated data generation.

**Self-Guided Complexity.** Instruction-following data should possess sufficient complexity to foster function-calling skills. LLMs learn more effectively when the complexity of the data slightly exceeds their current capability Du et al. (2023). To address this, we propose a self-guided dialog generation process (SDG), where the LLM serves as an evaluator to regulate complexity. Four types of function-calling data are generated with multi-agent interactions, following a self-guided complication strategy.

**Refined Accuracy.** Data accuracy is fundamental to the effectiveness of tool-augmented LLMs. ToolACE employs a dual-layer verification (DLV) system, integrating both rule-based and model-based checkers, to guarantee the executability and consistency of the synthesized data.

Equipped with data accuracy, complexity, and diversity, ToolACE aims to enhance the function-calling capability of LLMs with strong generalization. Our contributions are outlined as follows:

- We propose a novel automated data pipeline for function calls, ToolACE, which comprises a tool self-evolution synthesis module, a self-guided dialog generation module, and a dual-layer verification module. To our knowledge, this is the first work to highlight the benefits of synthesizing diverse APIs to improve the generalization of function calls.

- We develop a self-guided complication strategy to generate various types of function-calling dialogs with appropriate complexity. The given LLM is utilized as the complexity evaluator to guide the complexity level of the generated data. The quality of the generated data is ensured through a dual-layer verification process, which combines both rule checkers and model checkers.

- We conduct experiments on two widely adopted benchmarks: BFCL Yan et al. (2024) and APIBank Li et al. (2023). With only 8B parameters, ToolACE significantly outperforms existing open-source LLMs and is competitive with the latest GPT-4 models.

## 2 DATA GENERATION PIPELINE

Effective use of synthetic data significantly enhances the capabilities of large language models (LLMs) Mitra et al. (2024). Hence, in ToolACE, we propose an automated agentic framework for tool learning to generate high-quality, diverse, and complex data, guided by the capability of

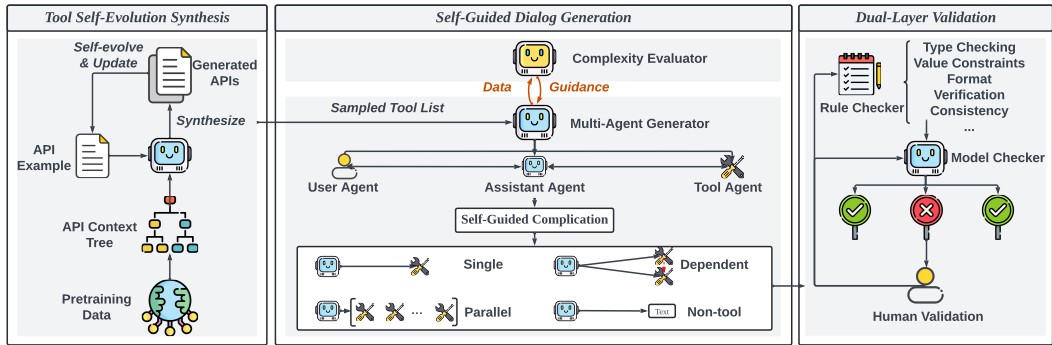

Figure 1: The overall framework of ToolACE, which mainly consists of Tool Self-evolution Synthesis (TSS), Self-Guided Dialog Generation (SDG), and Dual-Layer Validation Process (DLV).

the given LLM to be tuned, as illustrated in Figure 1. The proposed framework deploys various agents to recursively synthesize diverse APIs, collaboratively construct dialogs with appropriate complexity, and rigorously reflect on data quality. The following sections present our Tool Self-evolution Synthesis (TSS) module, Self-Guided Dialog Generation (SDG) module, and Dual-Layer Validation Process (DLV).

## 2.1 TOOL SELF-EVOLUTION SYNTHESIS

The variety of APIs significantly underpins the diversity of the function-calling data. As shown in Table 1, ToolACE has established a comprehensive API pool that surpasses other representative tool-augmented LLMs in both quantity and domain coverage, incorporating both real and synthesized APIs. Beyond collecting real API data, we developed a Tool Self-Evolution Synthesis (TSS) module that synthesizes API definitions with various data types and constraints, which encompasses three major steps: 1) Speciation, 2) Adaptation, and 3) Evolution.

**Speciation.** APIs with extensive domain coverage enable tool-augmented LLMs to learn a wider array of use cases from various applications and industries, thereby significantly enhancing their generalization ability. In the speciation step, we propose to create a hierarchical API context tree to guide the synthesis process with possible API domains and functionalities.

We observe that the pretraining data for LLMs encompasses one of the most diverse sources of human corpus, providing a solid foundation for extracting various API domains and use cases. Starting with API-related raw documents from the pretraining data (e.g., technical manuals, API documentation, product specifications, user guides, and tutorials), we prompt an agent powered by a frontier LLM to extract an API domain along with all possible API functionalities or use cases from each document. Children nodes of the context tree are recursively generated at each step, with each node denoting a possible API functionality (*e.g.*, get the weather forecast, get the stock price, send an email). Figure 9 in the Appendix A showcases the subtree under the *entertainment* domain as an example.

**Adaptation.** In the adaption step, we specify the domain and diversity level of each API. We sample a subtree and obtain unique functionalities from the API context tree for each individual API, so that different APIs possess distinct functionalities. For example, some APIs may cover more nodes, thereby acquiring more domain-specific and detailed capabilities. Whereas some APIs may only include a single node from the context tree, focusing on an easy, straightforward purpose.

**Evolution.** The evolution step involves the continuous improvement and adaptation of the API based on outcomes and new requirements. An LLM is instructed to synthesize new APIs according to a sampled subtree of the API context tree and an API example. The generated definitions of new APIs are required to be clear and thorough. We then apply a set of diversity indicators, *e.g.*, adding new functionalities or parameters, including additional constraints, mutating parameter type, and updating returned results, to diversify the generated APIs. We maintain an API example buffer containing various API examples. Iteratively, we sample an example from the buffer, adapt it to the current subtree of functionalities, and generate the next generation of the APIs.

The proposed TSS module facilitates the efficient generation of a diverse set of API documentation, with nested types including lists of lists or lists of dictionaries.

## 2.2 SELF-GUIDED DIALOG GENERATION

The effectiveness of function-calling data is closely tied to the capabilities of the LLM. For different LLMs, the knowledge and abilities they have learned during the pretraining phase are different, thereby the function-calling data they require should also differ Du et al. (2023). For instance, an LLM with 0.5B parameters may struggle to comprehend extremely complex data with long dependencies between APIs. In contrast, a well-trained 70B LLM can easily handle straightforward queries with clear intentions and simple APIs. In both cases, the data is unproductive for the given LLM, highlighting the importance of tailoring data generation to align with the model's capabilities.

Hence, to ensure the generated dialogs indeed fill the ability gap for the given LLM, we propose a self-guided dialog generation (SDG) module to synthesize the function-calling dialogs, as shown in the middle part of Figure 1. SDG consists of a complexity evaluator and a multi-agent generator. Various types of function-calling dialogs are generated via the interaction of multiple agents. The LLM to be tuned serves as the evaluator, assessing the complexity of the generated data. Data that is deemed too simple or too complex is dynamically adjusted under the guidance of the evaluator.

### 2.2.1 MULTI-AGENT DIALOG GENERATION

We propose a multi-agent framework to generate the four types of function-calling dialogs: single function calls, parallel function calls, dependent function calls, and non-tool-use dialogs.

The data generator includes three agents—user, assistant, and tool—each simulated by an LLM. One or more API candidates are sampled from our curated API pool and present the sampled APIs to the agents. Dialogs are then generated through role-playing among the three agents, each agent is provided with a necessary role assignment and detailed task description to continue the conversation. The *user agent* mainly makes requests or provides additional information to the assistant, with a self-guided complication process to adjust the dialog complexity. The *assistant agent* addresses the user's queries equipped with the given APIs. The action space of the assistant agent includes: calling the APIs, requesting further information, summarizing the tool feedback, and providing non-tool-use answers. To ensure data quality, each assistant action is generated multiple times, and only responses with consistent decisions across multiple instances are adopted. A specialized and structured thinking process specifically designed for function calls is also applied to enhance the assistant's tool-calling decisions. The *tool agent* acts as the API executor, processing tool descriptions and input parameters provided by the assistant, and outputs the potential execution results.

For each function-calling dialog, the user agent initiates a request related to the given sampled APIs. The assistant agent reviews the request and decides whether to call an API or ask for additional information. If tool calls are required, the tool agent will provide simulated results, and the assistant agent will summarize the results and present the user. The generation process continues with the user agent querying again or responding to the assistant's question until the target turn length is reached.

### 2.2.2 DATA COMPLEXITY EVALUATION

Different LLMs exhibit varying knowledge and capabilities, which necessitates the use of different data to optimize tool usage performance. However, much of the existing research overlooks the correlation between the model capability and the training data, leading to suboptimal data efficiency.

In this work, we employ the LLM to be tuned, denoted as $\mathcal{M}$, as the evaluator, and use the loss on a data sample of $(x, y)$ pairs for $\mathcal{M}$ to assess data complexity, denoted as $H_{\mathcal{M}}(x, y)$. The data complexity is measured as:

$$H_{\mathcal{M}}(x, y) = -\frac{1}{n_y} \sum_{i=1}^{n_y} \log p(t_i | x, t_1, \ldots, t_{i-1}), \tag{1}$$

where $x$ is the input query, and $y = [t_1, \ldots, t_{n_y}]$ is the response with $n_y$ tokens. Here, $t_i$ denotes the $i$-th token for $i = 1, \ldots, n_y$, and $p$ represents the probability of predicting the next token. A higher loss implies that the data sample $(x, y)$ has been found harder to learn for the model $\mathcal{M}$.

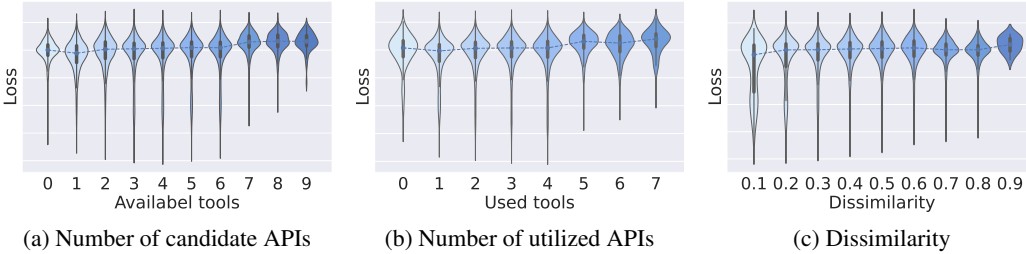

| (a) Number of candidate APIs | (b) Number of utilized APIs | (c) Dissimilarity |

Figure 2: Relationships between loss and (1) the number of candidate APIs available for selection, (2) the number of APIs utilized, and (3) the dissimilarity between the user query and the API descriptions.

Our findings suggest that the loss of a data sample is generally positively correlated with (1) the number of candidate APIs available for selection, (2) the number of APIs utilized, and (3) the dissimilarity between the user query and the API descriptions, as demonstrated in Figure 2. Intuitively, as the number of candidate APIs increases, selecting the correct one becomes more difficult. Similarly, the use of a higher number of APIs reflects greater query complexity, while larger discrepancies between the user query and the API descriptions demand more sophisticated reasoning to identify the correct function. These validate the use of loss as a measure of data complexity in function calling.

To establish an appropriate complexity range for the given LLM $\mathcal{M}$, we create a small, prior data set that spans various levels of complexity. A data sample that is correctly generated by $\mathcal{M}$ indicates that the model has already mastered the corresponding tool usage case, and thus this sample is unnecessary for further fine-tuning. The associated loss serves as a reference lower bound for data complexity. Conversely, if the loss of a data sample remains high after fine-tuning, it may indicate that the sample is too complex for the model to learn, and this loss serves as a reference upper bound.

Our evaluator provides the suitable complexity range, along with the loss of the given data sample, as the guidance information for the multi-agent generator in generating the training data.

### 2.2.3 SELF-GUIDED COMPLICATION

After obtaining the complexity of the current data from the evaluator, the user agent's instructions are dynamically adjusted. If the data sample is too simple for the LLM, the user agent is instructed to generate a more complex query–one that either requires additional APIs or diverges further from the API description to increase complexity. Conversely, if the data sample exceeds the LLM's capacity, the user agent is prompted to produce a simpler query. In this way, the data generation process is continually adapted to better match the model's performance level.

### 2.3 DUAL-LAYER DATA VERIFICATION

A critical factor influencing the function-calling capability of LLMs is the accuracy and reliability of the training data. Data that is inconsistent or inaccurate can hinder the model's ability to interpret and execute functions Liu et al. (2024). Unlike general question-answering data, where verifying correctness can be challenging, function-calling data is more verifiable. This is because a successful function call must strictly match the format specified in the API definition. Building on this insight, we propose an automatic dual-layer verification system (DLV) to verify our synthesized data, as shown in the right part of Figure 1, which consists of *a rule verification layer*, and *a model verification layer*, where these results are all overseen by human experts.

**Rule Verification Layer.** The rule verification layer deploys a rule checker to ensure that the data strictly adheres to the predefined syntactic and structural requirements of the API, covering four key aspects: API definition clarity, function calling executability, dialog correctness, and data sample consistency, guided by a meticulously curated set of rules, as listed in Appendix B.

For instance, to verify function calling executability, we implement the following procedures: First, we confirm that the API name matches one from the given tool list. Next, we verify that all required parameters are accurately provided. Finally, we use regular expressions to ensure that the parameter

formats and patterns adhere to those specified in the API documentation. These procedures allow us to validate the correctness and executability of function calls without the need for actual execution, which enhances efficiency and reduces deployment overhead.

**Model Verification Layer.** The model verification layer further incorporates LLMs to filter out erroneous data that cannot be detected by the rule checker, with a primary focus on content quality. However, we find that presenting a data sample directly to the LLM for correctness evaluation is too complex, often resulting in unsatisfactory outcomes. To address this, we decompose the model verification task into several sub-queries that mainly cover three key aspects:

- *Hallucination Detection:* Identifies whether the values of input parameters in function calls are fabricated—not mentioned in either the user query or the system prompt.
- *Consistency Validation:* Verifies that the responses can effectively complete the user's task and ensures the dialogue content adheres to the constraints and instructions in the user query and system prompt.
- *Tool Response Check:* Ensures that the simulated tool responses align with the API definition.

Each aspect is evaluated by an individual expert agent, powered by an LLM. We also incorporate other verification prompts to eliminate repetitive responses and meaningless tokens in the data.

## 3 EXPERIMENT

### 3.1 EXPERIMENT SETUP

To validate the effectiveness of our approach, we have conducted extensive experiments by training LLMs with the generated data. We train the open-source LLM, LLaMA3.1-8B-Instruct AI@Meta (2024), in the supervised fine-tuning (SFT) manner, for most of the experiments. We refer to the model as ToolACE-8B. We also validate our data with other backbone LLMs like Qwen-series Bai et al. (2023). Due to the limited resources, we adopt the parameter-efficient training strategy LoRA Hu et al. (2022) to fine-tune the model. As for the hyper-parameters setting, we adopt one of the most common settings, which sets the rank as 16 and alpha as 32 for all modules. We compare the overall performance with the state-of-the-art API-based and open-source models, like GPT-series [2], as well as fine-tuned function calling models including Gorilla-OpenFunctions-v2 Patil et al. (2023) and xLAM-series Liu et al. (2024). Experiments are conducted on two representative benchmarks, including **BFCL** Yan et al. (2024) [3] and **API-Bank** Li et al. (2023). The two benchmarks are comprehensive and executable function call evaluations specifically designed to assess the ability of LLMs to invoke functions. We then conduct in-depth ablation study to reveal the effectiveness of accuracy, diversity, and complexity. More experimental settings including benchmark details, evaluation metrics, and training settings are shown in Appendix C.

### 3.2 OVERALL PERFORMANCE ANALYSIS

To assess the effectiveness of our ToolACE-8B model regarding its functional calling capabilities, we compare our ToolACE-8B model with various representative models. The results are summarized in Table 2 and Table 3, respectively.

The findings in BFCL indicate that API-based models demonstrate significant advantages over open-source models, such as the Claude series and the GPT-4 series. Open-source models fine-tuned for function calling, such as Functionary and xLAM, exhibit competitive performance, but still fall short of the leading models. Our ToolACE-8B model outperforms most API-based and open-source models in both the AST and Exec categories of BFCL, and continues to exhibit substantial advantages over all the open-source models in the context of API-Bank, demonstrating the effectiveness of our training data for functional calling. This is mainly attributed to our accurate, diverse, and complex synthesized data, which enhances the zero-shot function calling capability of the LLM. Additionally, ToolACE

---

[2] https://chatgpt.com
[3] The overall performance is evaluated on the latest BFCL-v3 and subsequent studies are evaluated on only non-live categories since there are more testing samples in these categories, showing more robust results.

Table 2: Accuracy performance comparison on BFCL-v3 leaderboard (updated on 09/20/2024). The top 20 models are listed for comparison. FC denotes the model is tailored for functional calling. (A) and (E) present AST and executable category, respectively. **Rel** and **Irrel** are abbreviations for relevance and irrelevance.

| Rank | Overall | Model | Non-live (A) | Single turn Non-live (E) | Live (A) | Multi turn Multi turn | Hallucination Rel | Irrel |
|---|---|---|---|---|---|---|---|---|
| 1 | 59.49 | GPT-4-turbo-2024-04-09 (FC) | 82.65 | 83.80 | 73.39 | 21.62 | 70.73 | 79.79 |
| 2 | 59.29 | GPT-4o-2024-08-06 (FC) | 85.52 | 82.96 | 71.79 | 21.25 | 63.41 | 82.91 |
| 3 | 59.22 | ToolACE-8B (FC) | 89.27 | 90.07 | 73.21 | 14.37 | 85.37 | 83.81 |
| 4 | 59.13 | xLAM-8x22b-r (FC) | 89.75 | 89.32 | 72.81 | 15.62 | 97.56 | 75.23 |
| 5 | 58.45 | GPT-4o-mini-2024-07-18 (FC) | 82.83 | 81.80 | 67.53 | 25.75 | 82.93 | 71.83 |
| 6 | 57.94 | xLAM-8x7b-r (FC) | 88.44 | 85.89 | 71.97 | 15.75 | 92.68 | 72.35 |
| 7 | 57.21 | GPT-4o-mini-2024-07-18 (Prompt) | 86.54 | 87.95 | 72.77 | 11.62 | 80.49 | 79.20 |
| 8 | 55.82 | mistral-large-2407 (FC) | 84.12 | 83.09 | 67.17 | 20.50 | 78.05 | 48.93 |
| 9 | 55.67 | GPT-4-turbo-2024-04-09 (Prompt) | 91.31 | 88.12 | 67.97 | 10.62 | 82.93 | 61.82 |
| 10 | 54.83 | Claude-3.5-Sonnet-20240620 (FC) | 70.35 | 66.34 | 71.39 | 23.50 | 63.41 | 75.91 |
| 11 | 53.66 | GPT-4o-2024-08-06 (Prompt) | 80.90 | 77.89 | 73.88 | 6.12 | 53.66 | 89.56 |
| 12 | 53.43 | GPT-4o1-mini-2024-09-12 (Prompt) | 75.48 | 76.86 | 71.17 | 11.00 | 46.34 | 88.07 |
| 13 | 53.01 | Gemini-1.5-Flash-Preview-0514 (FC) | 77.10 | 71.23 | 71.17 | 13.12 | 60.98 | 76.15 |
| 14 | 52.53 | Gemini-1.5-Pro-Preview-0514 (FC) | 75.54 | 77.46 | 69.26 | 10.87 | 60.98 | 80.56 |
| 15 | 51.93 | GPT-3.5-Turbo-0125 (FC) | 84.52 | 81.66 | 59.00 | 19.12 | 97.56 | 35.83 |
| 16 | 51.78 | FireFunction-v2 (FC) | 85.71 | 84.23 | 61.71 | 11.62 | 87.80 | 52.94 |
| 17 | 51.78 | Open-Mistral-Nemo-2407 (FC) | 80.98 | 81.46 | 61.44 | 14.25 | 65.85 | 59.14 |
| 18 | 51.45 | xLAM-7b-fc-r (FC) | 86.83 | 85.02 | 68.81 | 0.00 | 80.49 | 79.76 |
| 19 | 51.01 | Gorilla-OpenFunctions-v2 (FC) | 87.29 | 84.96 | 68.59 | 0.00 | 85.37 | 73.13 |
| 20 | 49.63 | Claude-3-Opus-20240229 (FC) | 58.40 | 63.16 | 70.50 | 15.62 | 73.17 | 76.40 |
| 21 | 49.55 | Meta-Llama-3-70B-Instruct (Prompt) | 87.21 | 87.41 | 63.39 | 1.12 | 92.68 | 50.63 |

Table 3: Accuracy performance comparison on API-Bank evaluation system. **Bold** values represent the highest performance for API-based and open-source models, respectively.

| | Model | Call | Retrieval+Call |
|---|---|---|---|
| API-based | gpt-3.5-turbo-0125 | 70.43 | **52.59** |
| | gpt-4-0613 | 75.94 | 48.89 |
| | gpt-4-turbo-2024-04-09 | 72.43 | 39.26 |
| | gpt-4o-mini-2024-07-18 | 74.69 | 45.93 |
| | gpt-4o-2024-05-13 | **76.19** | 42.96 |
| Open-source | Alpaca-7B | 24.06 | 5.19 |
| | ChatGLM-6B | 23.62 | 13.33 |
| | Lynx-7B | 49.87 | 30.37 |
| | xLAM-7b-fc-r | 32.83 | 21.48 |
| | LLaMA-3.1-8B-Instruct | 71.18 | 37.04 |
| | ToolACE-8B | **75.94** | **47.41** |

excels in mitigating hallucination, achieving impressive relevance and irrelevance scores of 85.37% and 83.81%, respectively. These results highlight its ability in maintaining an excellent balance between the two categories, unlike other models that either suffer from significant imbalance or underperform in both categories. ToolACE-8B also consistently and significantly outperforms xLAM-7b-fc-r, which is also fine-tuned for function calling with similar size, in all categories, providing compelling evidence of its superiority. Furthermore, our ToolACE-8B model shows consistent advantageous performance on API-Bank compared with all open-source models, demonstrating comparable performance with GPT-4-series models.

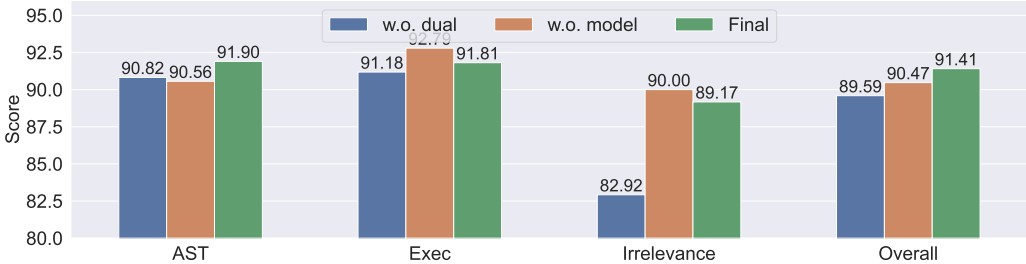

Figure 3: Ablation study of the dual-layer verification(DLV).

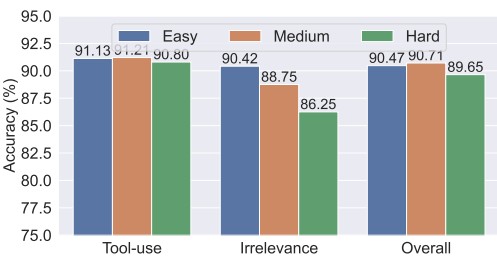

Figure 4: Ablation study on complexity.

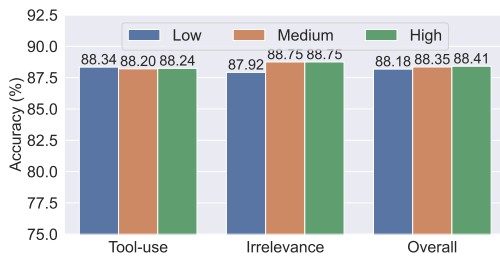

Figure 5: Ablation study on diversity.

## 3.3 ABLATION STUDY

### 3.3.1 ABLATION ON ACCURACY

***Effects of the verification system.*** As detailed in previous sections, our verification system comprises two layers: a rule checker and a model checker. To evaluate the efficacy of each layer, we train LLaMA3.1-8B-Instruct with LoRA using three distinct datasets: (1) data without any verification (denoted as **w.o. dual**), (2) data without model checking (denoted as **w.o. model**), and (3) data subjected to dual-layer verification (denoted as **Final**). It is important to note that datasets with more verification layers contain smaller amounts of data, as some data is filtered out during the verification process. The resulting fine-tuned models are assessed using the BFCL benchmark, with outcomes summarized in Figure 3. Comparative analysis reveals that the model trained on data without model checking surpasses that trained on unverified data in terms of both executable and overall accuracy, thereby validating the rule checker's effectiveness. Moreover, the model trained on dually verified data significantly outperforms both ablation models in terms of AST and overall accuracy, underscoring the indispensable role of the model checker.

### 3.3.2 ABLATION ON COMPLEXITY

***Data Sampling for Various Complexity.*** To effectively assess the impact of dataset complexity on the model's performance, we have conducted a sampling of the entire dataset based on the aforementioned complexity assessment metrics. We compute and sort the complexity for each data sample using Eq. (1), and select the bottom, middle, and top 60,000 instancess as ToolACE$_{easy}$, ToolACE$_{medium}$, ToolACE$_{hard}$, respectively, yielding three distinct subsets of varying complexity levels

The rationale behind this stratified sampling approach is to create a controlled environment where the influence of complexity can be systematically analyzed. By maintaining equal sample sizes across subsets, we ensure a fair comparison while varying the complexity, which allows for a more nuanced understanding of how complexity affects model performance.

***Effects of Complexity.*** We conduct experiments by training LLaMA-3.1-8B-Instruct with those three subsets with varying complexity and evaluate the fine-tuned models on the BFCL benchmark. The results are illustrated in Figure 4. The model trained on ToolACE$_{medium}$ shows slight superiority compared with another two subsets, for both overall and tool-use accuracy. This finding aligns with our hypothesis that optimal data complexity is essential for LLM training, as data that is either too simple or overly complex can prevent the model from reaching its full performance potential.

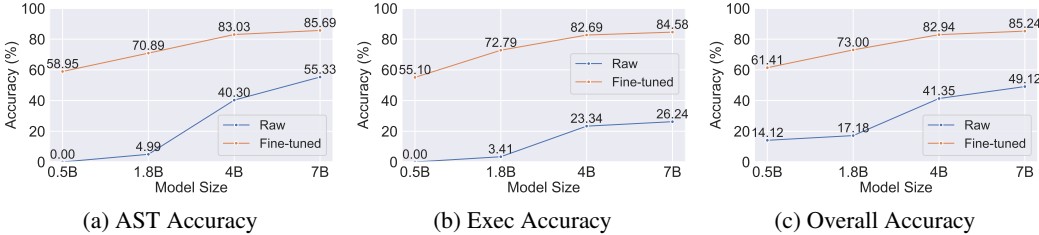

(a) AST Accuracy          (b) Exec Accuracy          (c) Overall Accuracy

Figure 6: Scaling performance of model size. The backbone LLMs are Qwen-1.5-xB-Chat series because this series offers models ranging from 0.5B to several billion parameters, enabling a comprehensive analysis of the relationship between model scale and performance.

### 3.3.3 Ablation on Diversity

***Data Sampling for Various Diversity.*** To assess the impacts of the diversity, we generate three subsets with varying diversity, namely ToolACE$_{low}$, ToolACE$_{medium}$, and ToolACE$_{high}$. Initially, all APIs are clustered into 30 groups based on the API context tree. Subsequently, three API subsets are constructed by selecting APIs from 6, 14, and 30 clusters, respectively. Instances are then categorized into three subsets according to their associated APIs. Approximately 30,000 instances are randomly selected from each subset, resulting in three training sets with distinct levels of diversity.

***Effects of Diversity.*** Experiments are conducted to train LLaMA-3.1-8B-Instruct on three subsets described above. The results on BFCL are reported in Figure 5. A positive correlation between training data diversity and overall model accuracy is observed, emphasizing the critical role of API diversity in model performance. Notably, improvements in relevance detection are particularly pronounced, suggesting that exposure to a wider range of APIs enhances the model's ability to discriminate between subtle API differences, thereby enhancing the ability of irrelevance detection.

### 3.4 Scaling Performance of Model Size

Scaling laws posit a correlation between model size and performance. To investigate the scalability of functional calling capabilities, we conduct experiments using the Qwen-1.5-xB-Chat series, which includes a range of model sizes (0.5B, 1.8B, 4B, 7B, etc.). Both raw and fine-tuned (using our dataset) models are evaluated on the BFCL benchmark, with results presented in Figure 6. As expected, larger models exhibit superior performance in functional calling, as evidenced by improvements in both AST and Executable accuracy. While smaller raw models (0.5B and 1.8B) showed minimal function-calling ability, struggling to generate structured outputs, fine-tuning on the ToolACE dataset significantly enhanced their capabilities. The fine-tuned models exhibit consistent scaling performance, highlighting the potential of ToolACE to boost the performance of larger LLMs.

### 3.5 Study on Various Backbone LLMs

To investigate the influence of the LLM backbone, we experiment with several (approximately) 8B-scale models: Qwen1.5-7B-Chat Bai et al. (2023), LLaMA-3-8B-Instruct, and LLaMA-3.1-8B-Instruct. Fine-tuned models are evaluated on the BFCL benchmark, with results presented in Figure 7. Across all models, fine-tuning yields substantial performance gains, highlighting the effectiveness of our ToolACE. Due to differences in pre-training corpora, such as Qwen is trained with more Chinese conversational samples, raw models exhibit varying functional calling capabilities, with LLaMA-3.1-8B-Instruct demonstrating superior performance. While this hierarchy persisted after fine-tuning, the performance gaps narrowed, suggesting that our dataset can potentially enhance the functional-calling abilities of those LLMs tailored for other skills, such as conversational skills.

### 3.6 Study on General Capabilities

To assess the impact of ToolACE training on broader capabilities of LLMs, we conduct experiments across multiple benchmarks evaluating general ability, including MMLU Hendrycks et al. (2021a;b), HumanEval Chen et al. (2021) (coding), GSM8K Cobbe et al. (2021) (mathematics), Common-

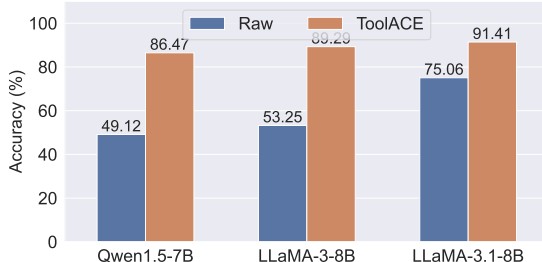
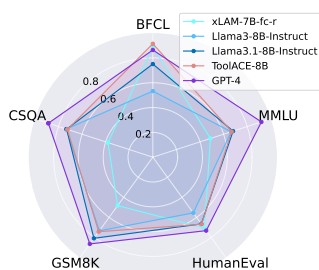

Figure 7: Performance on various LLMs.      Figure 8: General capabilities.

SenseQA Talmor et al. (2019) (reasoning), and BFCL Yan et al. (2024) (functional calling). Raw LLaMA-3-8B-Instruct, LLaMA-3.1-8B-Instruct, functionally specialized xLAM-7B-fc-r, and GPT-4 serve as baselines. Results are presented in Figure 8. ToolACE-8B substantially improves over xLAM-7B-fc-r across most benchmarks, with particularly pronounced gains in MMLU, GSM8K, and CommonSenseQA. Compared to GPT-4, ToolACE-8B shows clear limitations in reasoning and understanding. This is primarily due to the scale of the model and its training corpus. Compared to the raw LLaMA-3.1-8B-Instruct, ToolACE-8B demonstrates negligible performance degradation on some benchmarks while achieving significant enhancements in functional calling. These findings suggest that the ToolACE dataset effectively enhances functional calling capabilities without compromising the underlying LLM's general abilities. This success highlights the potential of specialized models in one specific domain, the challenge of simultaneously enhancing multiple capabilities, alongside functional-calling performance, remains an open question. The detailed analysis of the limitations can be referred to in Appendix H.

## 4 RELATED WORK

**Tool Learning.** Integrating external tools allows LLMs to expand the boundaries of their capabilities, enabling more specialized, precise, and dependable problem-solving (Qin et al., 2023). Methods for equipping LLMs with tool-use capabilities generally fall into two types: tuning-free approaches and tool-augmented tuning. Tuning-free methods let LLMs use tools by providing in-context tool descriptions and examples, requiring no additional training Mialon et al. (2023); Hsieh et al. (2023); Ruan et al. (2023). A well-known technique is ReAct Yao et al. (2023), which enables LLMs to alternate between reasoning and actions to solve complex tasks. However, as these approaches depend heavily on the model's initial abilities, tool-augmented tuning has gained more attention for directly improving tool use Qin et al. (2023); Schick et al. (2023); Patil et al. (2023); Tang et al. (2023); Liu et al. (2024); Abdelaziz et al. (2024). Many of these methods rely on existing APIs but lack robust systems for generating and validating data. Our ToolACE overcomes this limitation by implementing a well-designed pipeline that ensures greater diversity, complexity, and accuracy.

**Data Synthesis.** As LLMs grow more advanced, relying solely on existing human-generated data becomes insufficient for further progress Bauer et al. (2024). A key strategy involves modifying or augmenting datasets using specialized prompting techniques Wang et al. (2023); Xu et al. (2023); Yu et al. (2023). Given the scarcity of tool-use datasets, Basu et al. (2024) repurpose data from other domains for tool-use applications, while others Qin et al. (2023); Tang et al. (2023); Liu et al. (2024) depend on publicly available APIs, often producing single-turn instructions with basic tool interactions. ToolACE offers a more comprehensive approach, incorporating both tool synthesis and dialogue generation, along with a verification module to ensure data quality.

## 5 CONCLUSION

This paper presents ToolACE, an automated data generation pipeline developed to enhance the function-calling capabilities of large language models. ToolACE employs a novel self-evolution synthesis process and a self-guided data generation method to curate accurate, complex, and diverse synthetic APIs and dialogs. Our results demonstrate that even smaller models trained with ToolACE can achieve state-of-the-art performance, thereby advancing the field and setting new benchmarks for tool-augmented AI agents.

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

## A   AN EXAMPLE SUBTREE OF THE API CONTEXT TREE FOR THE *Entertainment* DOMAIN.

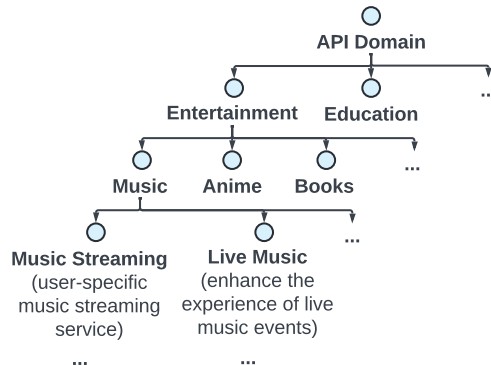

Figure 9: A subtree of the constructed API context tree for the *Entertainment* domain.

## B   RULE EXAMPLES IN RULE VERIFICATION LAYER

Table 4 outlines the check rules we use, which consists of four aspects: API definition clarity, function calling executability, dialog correctness, and data sample consistency.

Table 4: Example rules for the ToolACE rule checker.

| Aspect | Rules |
| --- | --- |
| API Definition Clarity | Check if the API definition complies with JSON Schema specifications. 
 Check if the API definition contains all necessary fields. |
| Function Calling Executability | Check if the API name is in the tool list. 
 Check if all required parameters are provided. 
 Check if all the parameter formats and patterns match the API definition. |
| Dialog Correctness | Check if the dialog contain all necessary fields. 
 Check if the assistant's response is too long. 
 Check for invalid characters in the responses. 
 Check for mixed-language responses. 
 Check if the response is complete. |
| Data Sample Consistency | Check if the API names in the function call and the tool response are consistent. 
 Check for format conflicts with the requirements defined in the system prompt. 
 Check if the order of the dialogue roles is correct. 
 Check if the tool response follows the function call. |

## C   EXPERIMENTAL DETAILS

### C.1   BENCHMARKS

**BFCL.**   Berkeley Function-Calling Benchmark (BFCL) Yan et al. (2024) is a comprehensive evaluation framework for assessing the function-calling capabilities of LLMs across various languages, application domains, and complex use cases. BFCL covers tasks including multiple function calls, parallel function calls, multi-turn function calls, and multi-step function calls. BFCL contains 4,951 test cases: 3,951 single-turn cases and 1,000 multi-turn cases, focusing on dynamic, real-world scenarios.

BFCL evaluates multiple function calling tasks using the following metrics:

- *Abstract Syntax Tree (AST) Evaluation:* AST evaluation compares the abstract syntax tree of the function output to the ground truth and the function definition. It captures the correctness of matching the functions, required parameters, parameter types, and values.

- *Executable Function Evaluation:* Executable function evaluation assesses the accuracy of the generated API call by executing it and comparing the output with the ground-truth output.

- *Irrelevance:* Irrelevance measures the model's ability to refrain from making function calls given irrelevant user queries. The irrelevance score is calculated as the number of correct non-function-call predictions divided by the total number of test samples.

- *Relevance:* Relevance evaluates the model's ability to output function calls relevant to the user query. In this category, the correctness of the parameter values is not considered. The relevance score is calculated as the number of correct function-call predictions divided by the total number of test samples.

- *Overall Accuracy:* Overall accuracy is the unweighted average of the accuracies across all sub-categories.

**API-Bank.** API-Bank Li et al. (2023) consists of 314 tool-use dialogues with 753 API calls to assess LLMs' capabilities in planning, retrieving, and calling APIs, with 363 single calls and 122 multiple calls. API-Bank assesses LLM performance across three capabilities:

- *Call:* The ability to call an API based on a given query when the APIs are known.

- *Retrieval+Call:* The ability to retrieve and call a single API when the APIs are unknown.

- *Plan+Retrieval+Call:* The ability to continuously plan, retrieve, and call multiple APIs when the APIs are unknown.

The evaluation metric for API-Bank is *accuracy*, calculated as the number of correct predictions divided by the total number of predictions.

## C.2 HYPER-PARAMETERS

The hyper-parameters of the training process are illustrated in Table 5.

Table 5: Hyper-parameters in experiments for training.

| Learning Rate | WarmUp Ratio | LR Scheduler | Batch Size | Epochs | LoRA rank | LoRA alpha |
|---|---|---|---|---|---|---|
| $10^{-4}$ | 0.1 | cosine | 48 | 3 | 16 | 32 |

## D CASE STUDY

Here we present a selection of cases from our generated data, showcasing various examples of tool utilization and function calls.

Figure 10 presents a data sample of parallel function calling. This type of data focuses on building the model's ability to understand and extract key parameters from the user query, which makes models learn to call the correct function repeatedly for accomplishing a task. In this example, the query indicates that the user needs to get the event information of Theatre, Dance, and Music during 2021-04-01 and 2021-05-01 respectively. The assistant correctly recognizes that it needs to call performanceArt.get_upcoming_events for three times with different assignments of the parameter "category".

Figure 11 shows a data sample of multiple function calling. This kind of data focuses on giving the model the ability to distinguish between different functions, which makes models learn to choose the correct function(s) for finishing a task. There are five functions in the tool list in the case. The user needs the information of the live event count for soccer, NBA stats for the 2025 season, and the latest

| Case 1: parallel function calling: | |
|---|---|
| ToolList | **performanceArt.get_upcoming_events** required: ["start_date", "end_date"] optional: ["category"] |
| User | I would like to know about upcoming Theatre, Dance, and Music performance art events happening between 2021-04-01 and 2021-05-01. |
| Assistant | **performanceArt.get_upcoming_events**(start_date="2021-04-01", end_date="2021-05-01", category="Theatre"), 
 **performanceArt.get_upcoming_events**(start_date="2021-04-01", end_date="2021-05-01", category="Dance"), 
 **performanceArt.get_upcoming_events**(start_date="2021-04-01", end_date="2021-05-01", category="Music") |

Figure 10: A case on parallel function calling.

| Case 2: multiple function calling: | |
|---|---|
| ToolList | **Bet365 Inplay Filter** required: optional: ["sport_id"] ["skip_esports"] 
 **Get Live Events Count by Sport** required: ["sport", "locale", "timezone"] 
 **National Team Statistics** required: ["player_id"] 
 **Get All NBA Stats** required: ["seasons"] optional: ["page", "per_page", "player_ids", "game_ids"] 
 **Get Latest Team Media** required: ["team_id"] |
| User | I plan to watch some sports events and want to catch up on the latest NBA stats. Could you find the live event count for soccer, get NBA stats for the 2025 season, and the latest media for the Lakers? |
| Assistant | **Get Live Events Count by Sport**(sport="soccer", locale="en_US", timezone=-4.0), 
 **Get All NBA Stats**(seasons=["2025"], page=1, per_page=10), 
 **Get Latest Team Media**(team_id=13.0) |

Figure 11: A case on multiple function calling.

media for the Lakers. The assistant chooses the corresponding Get Live Events Count by Sport, Get All NBA Stats, and Get Latest Team Media from the tool list and calls them to finish the task.

Figure 12 presents cases that focus on enhancing the model's ability to determine the relevance of a function to a user's query. In the first scenario, the user requests movie showtimes at specific cinemas. However, the available functions are designed only for retrieving event details or specific book characters, rendering them irrelevant to the user's request. The assistant appropriately declines to provide an answer, citing the lack of relevant functionality. In the second scenario, the user asks to start an Android emulator, but the essential parameters—device name, system image, and API level—are missing. The assistant correctly identifies this gap and requests the necessary information rather than attempting an incomplete operation. This type of data is vital for training the model to recognize when a function is not relevant or when it lacks the required inputs, ensuring that the system responds accurately and avoids executing unsupported tasks.

# E ADDITIONAL EXPERIMENTS

## E.1 COMPARISON WITH OTHER TRAINING DATA

To further highlight the value of our ToolACE dataset, we conducted additional experiments on two representative datasets (ToolLLM and xLAM), as summarized in Table 6. Specifically, we trained models using the amount of data (25,000 samples) and the same base model (LLaMA-3.1-8B-Instruct) to ensure a fair comparison. The trained models were then evaluated on the BFCL benchmark. The results show that the model trained with our dataset consistently outperforms the others across all

| Case 3: relevance | |
|---|---|
| ToolList | **Event** optional: ["name", "club", "club_name", "start_time", "end_time", "time_modified", "string_filter"]
**Get Character by ID** required: ["id"] |
| User | Can you check the showtimes for the film "Avengers: Endgame" at Cinema 21 and Cinema 45? |
| Assistant | The provided functions do not support searching for showtimes at specific cinemas. The functions available are limited to retrieving events based on filters or retrieving a specific character from the Throne of Glass series. Therefore, the question lacks the parameters required by the given functions. |
| ToolList | **runAndroidEmulator** required: [device_name, system_image, api_level] |
| User | Can you start the Android emulator for me? |
| Assistant | The function runAndroidEmulator can start the Android emulator, but it lacks the required parameters: device_name, system_image, and api_level. Please provide these details to proceed. |

Figure 12: A case on irrelevance detection.

categories, further validating the effectiveness of our approach. Notably, the model trained on the xLAM dataset exhibits relatively poor performance in irrelevance detection, likely due to a lack of diverse sample types, such as cases where provided tools cannot solve the task. Moreover, the ToolLLM dataset, which primarily focuses on multi-step and dependent cases, demonstrates weak generalization on the BFCL benchmark.

Table 6: Performances of training with different training datasets. The models are evaluated on the BFCL benchmark.

| Training data | Overall | Non-live(A) | Non-live(E) | Live(A) | Multi turn | Rel | Irrel |
|---|---|---|---|---|---|---|---|
| ToolLLM(2.5w) | 24.90 | 42.46 | 36.36 | 39.45 | 0.00 | 100.00 | 4.41 |
| xLAM(2.5w) | 40.51 | 81.94 | 81.77 | 43.18 | 4.38 | 73.17 | 11.87 |
| ToolACE(2.5w) (Ours) | 58.19 | 86.96 | 84.73 | 71.35 | 16.50 | 75.61 | 86.42 |

### E.2 ABLATION ON VARIOUS TYPES OF DATA

To underscore the importance of incorporating diverse data types—such as Nested, Parallel, Dependent, and Multi-type, as described in Table 1—we maintain the same overall dataset size (25,000) and selectively replace samples from the Nested, Parallel, Dependent, and Multi-type categories with samples from other data types. Then we train the LLaMA-3.1-8B-Instruct model and evaluate its performance on the BFCL benchmark. The results are summarized in Table 7.

The findings show that removing parallel execution data significantly impairs the model's ability to invoke multiple tools concurrently. This leads to a notable decrease in performance on Non-live AST and execution tasks, which rely heavily on parallel tool usage. Furthermore, excluding multi-type samples hampers the model's ability to detect when the candidate tools are irrelevant to the question, resulting in only 6.99% accuracy in irrelevance detection. The model's ability to handle multi-turn function calls is also impaired. In multi-turn testing, the models sometimes are required not to call functions, but to ask clarifying questions instead.

In contrast, removing nested and dependent samples has a relatively minor effect on the model's tool-using ability in the BFCL task. Few test samples require nested arguments, and almost none involve dependent tool usage. However, including Dependent and Nested data types contributes to greater data diversity, leading to slight improvements in overall performance.

Table 7: Ablation study on various types of data in ToolACE datasets. The models are evaluated on BFCL benchmark.

| Subset | Overall | Non-live(A) | Non-live(E) | Live(A) | Multi turn | Rel | Irrel |
|---|---|---|---|---|---|---|---|
| w.o. Parallel | 50.60 | 74.75 | 77.30 | 72.19 | 1.75 | 78.05 | 85.05 |
| w.o. Dependent | 57.97 | 87.63 | 85.55 | 71.17 | 15.50 | 80.49 | 85.62 |
| w.o. Nested | 57.19 | 85.46 | 84.48 | 70.19 | 15.38 | 78.05 | 86.45 |
| w.o. Multi-type | 42.71 | 89.46 | 85.50 | 47.89 | 1.75 | 95.12 | 6.99 |
| ToolACE | 58.19 | 86.96 | 84.73 | 71.35 | 16.50 | 75.61 | 86.42 |

Table 8: Ablation study on complexity evaluator. The evaluator represents the model used to evaluate the complexity. The learner denotes the model to be trained. Qwen-7B, Qwen-14B, and LLaMA-8B are abbreviations of Qwen1.5-7B-Chat, Qwen1.5-14B-Chat, and LLaMA-3.1-8B, respectively.

| Evaluator | Learner | Overall | Non-live(A) | Non-live(E) | Live(A) | Multi turn | Rel | Irrel |
|---|---|---|---|---|---|---|---|---|
| Qwen-7B | LLaMA-8B | 57.61 | 90.42 | 85.88 | 71.30 | 13.12 | 87.80 | 78.12 |
| Qwen-14B | LLaMA-8B | 57.67 | 87.98 | 87.02 | 73.30 | 11.75 | 87.80 | 84.00 |
| LLaMA-8B | LLaMA-8B | 59.22 | 89.27 | 90.07 | 73.21 | 14.37 | 85.37 | 83.81 |

### E.3 ABLATION ON COMPLEXITY EVALUATOR

To assess the complexity of the training data, we propose a self-guided evaluation method, where the model being trained serves as its own evaluator. To verify the suitability of this approach, we conduct an additional experiment using an independent model (Qwen1.5-7B-Chat, selected for its comparable size to ensure fairness) as the evaluator. The results, shown in Table 8, indicate that using the model being trained as the complexity evaluator offers more accurate guidance, leading to improved performance on the BFCL benchmark. Notably, when the complexity score is assessed using a more advanced model (Qwen-14B), some simpler training samples—those deemed easy by the evaluator but not necessarily by the learner—may be excluded. This leads to slight performance gains on more challenging tasks (e.g., Live AST) but results in degradations on Non-live AST tasks [4]. Conversely, when the evaluator is less capable than the learner, the retained samples tend to be relatively easier for the learner, resulting in improved performance on Non-live AST tasks but a decline in performance on Live AST tasks.

Table 9: Comparison between in-context learning and finetuning.

| Method | Non-live(A) | Non-live(E) | Live(A) | Rel | Irrel |
|---|---|---|---|---|---|
| LLaMA-8B (3-shot) | 58.81 | 53.32 | 36.83 | 82.93 | 23.66 |
| ToolACE (finetuning) | 89.27 | 90.07 | 73.21 | 85.37 | 83.81 |

## F PROMPTING TEMPLATES

To provide a better comprehension of the two benchmarks used in experiments, we have illustrated two examples for BFCL and API-Bank in Figure 13 and Figure 14, respectively.

## G FINETUNING VS IN-CONTEXT LEARNING

Given 3 shots for LLaMA-3.1-8B-Instruct, the model still fails to generate correct arguments for such a simple example, such as Figure 16, demonstrating the limited ability in tool using under the in-context learning setting. Besides, due to the addition of few-shot examples, the input to the

---

[4]Live AST tasks involve rarer and more complex functions compared to Non-live AST tasks, as detailed in BFCL's documentation.

model consumes a lot more tokens than the fine-tuned model, which successfully addresses the aforementioned example in a zero-shot setting, as presented in Figure 15.

Furthermore, we conducted experiments on BFCL under the RAG-based few-shot in-context learning setting. Specifically, we use the training samples as few-shot examples and retrieve the top 3 most relevant ones according to the user's question and the provided tools with the BGE model to guide in-context learning. The results illustrated in Table 9 show that few-shot in-context learning not only underperforms fine-tuning in BFCL but also falls short of the zero-shot setting. In many cases, illustrated in Figure 16, the model is misled by the tools in the few-shot examples due to its limited reasoning ability and generalization, selecting those instead of the tools in the test sample, which further exacerbates the model's hallucination phenomenon.

## H    LIMITATIONS

While we have conducted extensive experiments to demonstrate the effectiveness of our synthesis dataset in enhancing the functional-calling performance, several challenges remain in our research.

- **Data Complexity Evaluation.** The computational complexity of data complexity evaluation is influenced by the size of the model being trained, which limits scalability as both the model size and the number of training samples increase. Additionally, the non-uniform sampling may introduce bias, such as causing the model to struggle with learning difficult examples after one round of training, effectively remaining in its *comfort zone*. In future work, we will further explore the proposed complexity-based sampling strategy to perform iterative training and sampling over multiple rounds, thereby progressively enhancing the model's generalization capability on more challenging samples.

- **General Capabilities.** Although ToolACE demonstrates comparable performance in functional calling, it still lags behind GPT-4 in other capabilities. While this success highlights the potential of specialized models in one specific domain, the challenge of simultaneously enhancing multiple capabilities, alongside functional-calling performance, remains an open question. Exploring the collaboration of multiple small, domain-specific agents may provide a promising direction for addressing this limitation.

```
System: You are an expert in composing functions. You are given a question and a set
of possible functions.
Based on the question, you will need to make one or more function/tool calls to
achieve the purpose. If none of the functions can be used, point it out. If the
given question lacks the parameters required by the function, also point it out. You
should only return the function call in the tools call sections.
If you decide to invoke any of the function(s), you MUST put it in the format
of [func_name1(params_name1=params_value1, params_name2=params_value2...),
func_name2(params)] You SHOULD NOT include any other text in the response.
Here is a list of functions in JSON format that you can invoke:

[{"name": "get_weather_data", "description": "Fetches weather data from the
Open-Meteo API for the given latitude and longitude.", "parameters": {"type":
"dict", "properties": {"coordinates": {"type": "array", "items": {"type":
"float"}, "description": "The latitude and longitude of the location."}}, "required":
["coordinates"]}}, {"name": "calc_binomial_probability", "description": "Calculates
the probability of getting k successes in n trials.", "parameters": {"type": "dict",
"properties": {"n": {"type": "integer", "description": "The number of trials."},
"k": {"type": "float", "description": "The number of successes."}, "p": {"type":
"float", "description": "The probability of success."}}, "required": ["n", "k",
"p"]}}]

User: I'm planning a small outdoor event in Ottawa, and I need to make sure the
weather is going to cooperate. Could you fetch the current weather for me at latitude
45.4215 and longitude -75.6972 using the Open-Meteo API? Also, I'm running a small game
at the event, and I'm curious about the chances of winning. If I have 10 attempts at
this game and the chance of winning each time is 50%, how likely is it that I'll win 5
times?

Assistant: [get_weather_data(coordinates=[45.4215, -75.6972]),
calc_binomial_probability(n=10, k=5.0, p=0.5)]
```

Figure 13: Example of BFCL benchmark. The blue text represents the part to be evaluated.

```
System:  Based on the given API description and the existing conversation history 1..t,
please generate the API request that the AI should call in step t+1 and output it in
the format of [ApiName(key1='value1', key2='value2', ...)], replace the ApiName with
the actual API name, and replace the key and value with the actual parameters.
Your output should start with a square bracket "[" and end with a square bracket "]".
Do not output any other explanation or prompt or the result of the API call in your
output.
This year is 2023.
Input:
User:  [User's plain text or response of API call]
AI: [AI's plain text]
...
User:  [User's plain text or response of API call]
Expected output:
[ApiName(key1='value1', key2='value2', ...)]
API descriptions:
[{"name":  "GetUserToken", "description":  "Get the user token by username and
password.", "input_parameters": {"username": {"type":  "str", "description":  "The
username of the user."}, "password": {"type":  "str", "description":  "The password
of the user."}}, "output_parameters": {"token": "type": "str", "description":
"The token of the user."}}, {"name":  "AddAlarm", "description":  "The API for
setting an alarm includes a parameter for the alarm time.", "input_parameters":
{"token": {"type":  "str", "description":  "User's token."}, "time": {"type":
"str", "description":  "The time for alarm.  Format:  %Y-%m-%d %H:%M:%S"}},
"output_parameters": {"status": {"type":  "str", "description":  "success or
failed"}}} ]

User:  Can you help me set an alarm for 8 am tomorrow?

Assistant:  Sure, to set an alarm, I need to authorize your account.  Can you please
provide your username and password?

User:  Sure, my username is foo and password is bar.

Assistant:  [GetUserToken(username="foo", passward="bar")]

Tool:  [GetUserToken Response: {"token":  "z9x8c7v6b5n4m3q2w1"}]

Assistant:  Okay, I got your token.  What time do you want the alarm to be set for?

User:  8 am tomorrow.  Today is 2021-10-13.

Assistant:  [AddAlarm(token="z9x8c7v6b5n4m3q2w1", time="2021-10-14 08:00:00")]

Tool:  [AddAlarm Response:  "success"]

Assistant:  An alarm has been set for 8 am tomorrow.
```

Figure 14: Example of API-Bank benchmark. The blue text represents the part to be evaluated.

```
System:  You are an expert in composing functions.  You are given a question and a
set of possible functions.  Based on the question, you will need to make one or more
function/tool calls to achieve the purpose.  If none of the function can be used, point
it out.  If the given question lacks the parameters required by the function, also
point it out.  You should only return the function call in tools call sections.

If you decide to invoke any of the function(s), you MUST put it in the format
of [func_name1(params_name1=params_value1, params_name2=params_value2,...),
func_name2(params)] You SHOULD NOT include any other text in the response.
Here is a list of functions in JSON format that you can invoke.  [{'name':
'calc_absolute_pressure', 'description':  'Calculates the absolute pressure from
gauge and atmospheric pressures.', 'parameters':  { 'type':  'dict', 'properties':
{ 'atm_pressure':  { 'type':  'integer', 'description':  'The atmospheric pressure in
atmospheres (atm).  Default is 1 atm if not provided.'  }, 'gauge_pressure':  { 'type':
'integer', 'description':  'The gauge pressure in atmospheres (atm).  Must be provided.'
} }, 'required':  [ 'gauge_pressure' ] }]

User:  Calculate the absolute pressure in pascals given atmospheric pressure of 1 atm
and a gauge pressure of 2 atm.

Assistant:  [calc_absolute_pressure(atm_pressure=1, gauge_pressure=2)]
```

Figure 15: Case: model fine-tuned with our ToolACE data can solve the question in the zero-shot setting.

```
System:  You are an expert in composing functions.  You are given a question and a
set of possible functions.  Based on the question, you will need to make one or more
function/tool calls to achieve the purpose.  If none of the function can be used, point
it out.  If the given question lacks the parameters required by the function, also
point it out.  You should only return the function call in tools call sections.

If you decide to invoke any of the function(s), you MUST put it in the format
of [func_name1(params_name1=params_value1, params_name2=params_value2,...),
func_name2(params)] You SHOULD NOT include any other text in the response.  Here is
a list of functions in JSON format that you can invoke.
[{"name": "solve_quadratic", "description": "Solve a quadratic equation given
coefficients a, b, and c.  If optional 'root_type' is 'real', the function will only
return real roots.  If not specified, function may return complex roots.", "parameters":
{"type": "dict", "properties": {"a": {"type": "integer", "description": "The
coefficient of the squared term in the quadratic equation."}, "b": {"type": "integer",
"description": "The coefficient of the linear term in the quadratic equation."}, "c":
{"type": "integer", "description": "The constant term in the quadratic equation."},
"root_type": {"type": "string", "description": "The type of roots to return: 'real'
for real roots, 'all' for both real and complex roots.  Default value is 'real'."}},
"required": ["a", "b", "c"]}}]

Here are some examples you can refer:
===
Available tools: [{'name': 'FunctionIntersect.calculateRoots', 'description':
'Identifies the roots of the equation formed by setting two functions equal to each
other.', 'parameters': {'type': 'dict', 'properties': {'equation': {'description':
"The equation obtained by setting two functions equal, e.g., '3x2 + 2x - 1 = x3
- 2x + 4'.", 'type': 'string'}, 'precision': {'description': 'The numerical
precision for calculating roots, specified as a decimal.', 'type': 'float'},
'method': {'description': "The numerical method to use for finding roots, such as
'Newton-Raphson' or 'Bisection'.", 'type': 'string'}}, 'required': ['equation']}}]
User:  Could you please help me find the roots of the equation 3x2 - 7x + 2 = 0
using the Bisection method and again using the Newton-Raphson method?  Assistant:
[FunctionIntersect.calculateRoots(equation="3x2 - 7x + 2 = 0", method="Bisection",
precision=0.01), FunctionIntersect.calculateRoots(equation="3x2 - 7x + 2 = 0",
method="Newton-Raphson", precision=0.01)]

Available tools: [{'name': 'complex_root_finder', 'description': 'Finds roots of a
complex polynomial within a specified region.', ...
User:  I need to find the roots of the polynomial "z4 + z3 + 2z2 + z + 6" within four
different regions specified by the following centers and radii:  (1+2i, 5), (-3+4i, 3),
(0-1i, 4), (2+2i, 6).  Each analysis should be done with high resolution during the
optimal times of day.
Assistant:  [complex_root_finder(polynomial="z4 + z3 + 2z2 + z + 6", region={"center":
{"real": 1.0, "imaginary": 2.0, "radius": 5.0}, search_parameters={"resolution":
1000, "time": "morning"}), complex_root_finder(polynomial="z4 + z3 + 2z2
+ z + 6", region={"center": {"real": -3.0, "imaginary": 4.0}, "radius":
3.0}, search_parameters={"resolution": 1000, "time": "morning"}),
complex_root_finder(polynomial="z4 + z3 + 2z2 + z + 6", region={"center": "real": 0.0,
"imaginary": -1.0, "radius": 4.0, search_parameters="resolution": 1000, "time":
"morning"), complex_root_finder(polynomial="z4 + z3 + 2z2 + z + 6", region={"center":
{"real": 2.0, "imaginary": 2.0}, "radius": 6.0}, search_parameters={"resolution":
1000, "time": "morning"})]

Available tools:  [{'name': 'calculus.integralSolver', 'description': 'Solves
definite and indefinite integrals for a given mathematical function.', 'parameters':
{'type': 'dict', 'properties': {'function': {'type': 'string', 'description':
'Mathematical function expressed in standard mathematical notation.'}, 'limits':
{'type': 'dict', 'properties': {'lower': {'type': 'string', 'pattern':
'^-?\\d+(\\.\\d+)?$', 'description': "Lower limit of the integral.  Must be a
numeric value, e.g., '-3.5', '0', '2'."}, 'upper': {'type': 'string', 'pattern':
'^-?\\d+(\\.\\d+)?$', 'description': "Upper limit of the integral.  Must be a numeric
value, e.g., '3', '4.5', '10'."}}, 'required': ['lower', 'upper']}}, 'required':
['function']}}]
User:  I'm currently preparing for my calculus exam and need to solve some integrals.
Can you solve the integral of the function f(x) = 3x2 from x = 0 to x = 4 for me?
Assistant:  [calculus.integralSolver(function="lambda x:  3*x**2", limits={"lower":
"0", "upper":  "4"})]

===
User:  Find all the roots of a quadratic equation given coefficients a = 3, b = -11,
and c = -4.

Assistant:  [calculus.integralSolver(function="lambda x:  3*x**2", limits="lower":  "0",
"upper":  "4")]
```

Figure 16: Case: Few-shot in-context learning cannot solve the quite easy question, hallucinating on calling tools in few-shot examples instead of the test sample.

