# OpenReview forum: "ToolACE: Winning the Points of LLM Function Calling"
_ICLR.cc/2025/Conference — ICLR 2025 Poster_

### Official Review · Reviewer_ygcU · 2024-10-27

**Soundness:** 3
**Presentation:** 2
**Contribution:** 4
**Rating:** 8
**Confidence:** 4

**Summary:**

ToolACE sampled API-related data from LLM pre-training corpora, obtaining 26,507 APIs, and used a User Agent - Assistant Agent - Tool Agent structure to synthesize appropriate conversational data to obtain API call training data. Rules + LLM were used during the process to ensure the effectiveness of the synthetic data. Finally, it used an 8B model with LoRA to validate the effects.

**Strengths:**

1. Ranks third on the BFCL-v3 leaderboard (updated on 09/20/2024), and first among open-source models on API-Bank.
2. The relatively large volume of synthetic data demonstrates the benefits of model fine-tuning at a larger scale.
3. Covers various categories of tool calls including Nested, Parallel, Dependent, and Multi-type.

**Weaknesses:**

1. The paper shows high complexity with many unclear details. For example, how are the varying complexity levels (easy, medium, hard) actually defined?
2. While the paper repeatedly mentions Nested, Parallel, Dependent, and Multi-type, it doesn't analyze their connection to actual performance or conduct ablation studies.
3. The relationship between data scale and performance is not demonstrated, making it difficult to determine which aspects actually contributed to the effectiveness.
4. There are concerns about potential data leakage between BFCL and TSS - can the authors prove there isn't significant data leakage?

**Questions:**

1. Since Nested, Parallel, Dependent, and Multi-type are essentially subsets of programming language capabilities, if they are indeed effective, does this suggest that direct training with programming languages (like Python) would be better? Furthermore, is the Data Interpreter[1] approach of directly exposing tool interfaces through Python a better solution? This needs further analysis.

[1] Hong, Sirui, et al. "Data interpreter: An LLM agent for data science." arXiv preprint arXiv:2402.18679 (2024).

---

> ### Author Response · Authors · 2024-11-17
> **Responses to Weakness 1, 2**
>
> **W1:** **The paper shows high complexity with many unclear details. For example, how are the varying complexity levels (easy, medium, hard) actually defined?**
>
> **A1:** Thank you for raising this point. The three levels of complexity (easy, medium, hard) are defined based on the complexity scores of the training samples, which are computed using Eq.(1). We first calculate the complexity of each sample and then sort all samples in ascending order of their complexity. The top 60,000 samples are classified as the "hard" subset, the bottom 60,000 as the "easy" subset, and the middle 60,000 as the "medium" subset. The detailed explanation has been included in our revised manuscript in Sect. 3.3.2, paragraph 1.
>
> Moreover, to make our manuscript clearer, we provide extended experimental details in
> C.1, and an example of the prompt and the response for the two benchmarks in Appendix F.
>
> **W2:** **While the paper repeatedly mentions Nested, Parallel, Dependent, and Multi-type, it doesn't analyze their connection to actual performance or conduct ablation studies.**
>
> **A2:** Thank you for your insightful comment. While we emphasize the significance of incorporating diverse data types such as Nested, Dependent, and Multi-type, there is currently no publicly available evaluation set that specifically addresses these categories. However, we acknowledge the importance of exploring the relationship between these data types and overall function-calling performance. We have conducted additional experiments in our revision in Appendix E.2. Specifically, we maintain the same overall dataset size and selectively replace samples from the Nested, Parallel, Dependent, and Multi-type categories with samples from other data types. We then train the Llama3.1-8B model using these modified subsets and evaluate its performance on the BFCL benchmark. The results are presented below.
>
> The findings show that removing parallel execution data significantly impairs the model's ability to invoke multiple tools concurrently. This leads to a notable decrease in performance on Non-live AST and execution tasks, which rely heavily on parallel tool usage. Furthermore, excluding multi-type samples hampers the model's ability to detect when the candidate tools are irrelevant to the question, resulting in only 6.99\% accuracy in irrelevance detection. The model's ability to handle multi-turn function calls is also impaired. In multi-turn testing, the models sometimes are required not to call functions but to ask clarifying questions instead.
>
> In contrast, removing nested and dependent samples has a relatively minor effect on the model's tool-using ability in the BFCL task. Few test samples require nested arguments, and almost none involve dependent tool usage. However, including Dependent and Nested data types contributes to greater data diversity, leading to slight improvements in overall performance.
>
> **Table 1. Ablation study on various types of data in ToolACE datasets.**
> | Data | Overall | Non-live(A) | Non-live(E) | Live(A) | Multi turn | Rel | Irrel|
> | -------- | -------- | -------- | -------- | -------- | -------- | -------- | -------- |
> | w.o. Parallel  | 50.60 | 74.75 | 77.30 | 72.19 | 1.75 | 78.05 | 85.05 |
> | w.o. Dependent | 57.97 | 87.63 | 85.55 | 71.17 | 15.50 | 80.49 | 85.62 |
> | w.o. Nested | 57.19 | 85.46 | 84.48 | 70.19 | 15.38 | 78.05 | 86.45 |
> | w.o. Multi-type | 42.71 | 89.46 | 85.50 | 47.89 | 1.75 | 95.12 | 06.99 |
> | ToolACE(2.5w)  | 58.19 | 86.96 | 84.73 | 71.35 | 16.50 | 75.61 | 86.42 |

---

> ### Author Response · Authors · 2024-11-17
> **Responses to Weakness 3, 4 and Question 1**
>
> **W3:** **The relationship between data scale and performance is not demonstrated, making it difficult to determine which aspects actually contributed to the effectiveness.**
>
> **A3:** Thanks for the valuable point. We recognize the importance of a more fair comparison and have conducted additional experiments under a controlled setting in revision in Appendix E.1. Specifically, we compare the performances of using ToolACE and other state-of-the-art function calling data (ToolLLM and xLAM) to train the same base model (Llama3.1-8B). All data are uniformly sampled to 25,000 for a fair comparison. The corresponding results on BFCL are presented in the table below. These results demonstrate that the model trained with our data consistently outperforms the other models in all categories, further validating the effectiveness of our approach. Notably, the model trained on the xLAM dataset exhibits relatively poor performance in irrelevance detection, likely due to a lack of diverse sample types, such as cases where provided tools cannot solve the task. Moreover, the ToolLLM dataset, which primarily focuses on multi-step and dependent cases, demonstrates weak generalization on the BFCL benchmark.
>
> **Table 2. Performances of training with different training datasets.**
> | Training data | Overall | Non-live(A) | Non-live(E) | Live(A) | Multi turn | Rel | Irrel|
> | -------- | -------- | -------- | -------- | -------- | -------- | -------- | -------- |
> | ToolLLM(2.5w) | 24.90 |  42.46 | 36.36 | 39.45 | 0.00 | 100.00 | 4.41 |
> | xLAM(2.5w)    | 40.51 | 81.94 | 81.77 | 43.18 | 4.38 | 73.17 | 11.87 |
> | ToolACE(2.5w) (Ours) | **58.19** | **86.96** | **84.73** | **71.35** | **16.50** | **75.61** | **86.42** |
>
> **W4:** **There are concerns about potential data leakage between BFCL and TSS - can the authors prove there isn't significant data leakage?**
>
> **A4:** We employ both the N-gram-based method and the similarity-based method to show that there is no significant data leakage in our ToolACE dataset.
>
> - N-gram-based method: Following the method used in Llama2, we consider a token to be contaminated if it appears in any token n-gram longer than 10 tokens in both the evaluation sample and the training set. A tool is classified as leaked if more than 10% of the tokens in its JSON string are contaminated. Under this setting, only a negligible percentage of 0.148% tools in the ToolACE dataset are leaked, compared to 0.610% in xLAM dataset.
> - Similarity-based method: We define a tool as leaked if the cosine similarity between the given tool and any tool in the evaluation dataset exceeds 0.9. We choose the BAAI/bge-large-en in huggingface as the encoder to get representations of all tools. Using this method, the proportion of leaked tools in our dataset is only 0.974%, compared to 5.214% in the xLAM dataset.
>
>
> **Q1:** **Since Nested, Parallel, Dependent, and Multi-type are essentially subsets of programming language capabilities, if they are indeed effective, does this suggest that direct training with programming languages (like Python) would be better? Furthermore, is the Data Interpreter approach of directly exposing tool interfaces through Python a better solution? This needs further analysis.**
>
> **A5:** We appreciate this insightful suggestion and agree that incorporating programming data like Python has the potential to enhance the model's function-calling capabilities. Previous research has similar ideas—for example, xLAM-7B uses DeepSeek-Coder-7B-instruct-v1.5 as its base model instead of a more general-purpose instruction model. While this paper focuses primarily on generating synthetic dialogue data for function calling, exploring the possibility of translating our synthetic data into Python code snippets would be an interesting and exciting research topic. We plan to investigate it in our future work.

---

> ### Comment · Reviewer_ygcU · 2024-11-20
>
> The reply didn't address all of my concerns, in particular, I still think the path seems to be directly to code, not json. But after a long period of thinking, I think this paper brings a new data synthesis strategy, although it may be a bit complicated, which has significant practical significance and is good for the field. I decided to raise my score.

---

> > ### Author Response · Authors · 2024-11-20
> > **Thank you**
> >
> > Thank you for re-evaluating our manuscript and for your thoughtful feedback. We appreciate your recognition of the significance of our data synthesis strategy and your consideration of its impact on the field. Your insights regarding the path to code have been highly inspiring, and we plan to delve deeper into this direction in our future work and hope to present more interesting results.

---

### Official Review · Reviewer_ViF9 · 2024-10-29

**Soundness:** 3
**Presentation:** 2
**Contribution:** 3
**Rating:** 6
**Confidence:** 2

**Summary:**

The paper "ToolACE: Enhancing Function Calling with Accuracy, Complexity, and Diversity" presents a novel data generation pipeline for function-calling tasks LLMs. The approach leverages a tool self-evolution synthesis module, a self-guided dialog generation module, and a dual-layer verification module to create accurate, complex, and diverse tool-calling scenarios. ToolACE aims to improve LLMs' zero-shot function-calling capabilities by generating comprehensive training data that is validated through rule-based and model-based checks. The experiments show promising results, particularly with the ToolACE-8B model, which outperforms several existing LLMs.

**Strengths:**

- The introduction of ToolACE's multi-step data generation, including evolutionary diversity and self-guided complexity, provides an innovative solution for generating complex and diverse function-calling data.

- The DLV system, combining rule-based and model-based checks, enhances the reliability of the generated data. This is a strong point, as it helps maintain data quality, which is critical for training LLMs effectively.

- The paper provides an extensive set of experiments, including comparisons with state-of-the-art models and an ablation study to assess the contribution of different components like accuracy, complexity, and diversity in the dataset. These experiments illustrate the potential benefits of the proposed pipeline.

**Weaknesses:**

- The evaluation scenarios are limited to synthetic function-calling tasks and benchmarks like BFCL and APIBank. The paper would benefit from more realistic evaluations or applications in real-world tool usage scenarios. This would better demonstrate ToolACE’s utility beyond controlled benchmark settings.

- The self-guided dialog generation process heavily relies on the LLM being trained to evaluate the complexity of generated data. This creates a circular dependency where the model is used both as a learner and an evaluator, which may introduce bias in the complexity estimation. More external validation or use of independent evaluators would make the results more robust.

- The use of complexity-based sampling to dynamically adjust dialog difficulty has merit but may lead to unintended biases, as data that is either too simple or too complex is filtered out. The approach may fail to fully explore the impact of diverse and extreme cases, leading to gaps in the model’s capabilities in certain contexts.

- While the paper compares ToolACE to several other function-calling models, the comparison is often superficial. The benefits of using ToolACE versus simpler data augmentation techniques are not well articulated, and it is unclear how much of the improvement can be attributed to the synthesis method versus the increased volume of data.

- The paper claims that ToolACE-8B is competitive with GPT-4 series models. However, it does not fully address the limitations of ToolACE-8B in terms of generalization and applicability to a broader range of tasks beyond function calling. A more detailed discussion of these limitations would provide a more balanced perspective.

- The font size in Figures is too small, which is unclear for readers.

**Questions:**

Please refer to weaknesses.

---

> ### Author Response · Authors · 2024-11-17
> **Responses to Weakness 1, 2, 3**
>
> **W1:** **The evaluation scenarios are limited to synthetic function-calling tasks and benchmarks like BFCL and APIBank. The paper would benefit from more realistic evaluations or applications in real-world tool usage scenarios. This would better demonstrate ToolACE’s utility beyond controlled benchmark settings.**
>
> **A1:** We thank the reviewer for the comments. Concerning this question, we would like to clarify that:
>
> 1. BFCL and APIBank are two widely adopted benchmarks to evaluate and compare LLMs' function-calling capability.
> 2. Most of the APIs in BFCL and APIBank are real APIs, and some of them can even be executed and evaluated by actually executing these APIs (Executable category in BFCL). Furthermore, the instances in the Live category of BFCL are collected from user-contributed function documentation and queries, "to more faithfully measure the LLM's function-calling performance in real-world scenarios" ([source](https://gorilla.cs.berkeley.edu/blogs/12_bfcl_v2_live.html)). Considering this, we believe the two benchmark evaluations can reflect ToolACE's utility in real-world scenarios, at least to some extent.
>
> Moreover, ToolACE has already been deployed to a real-world travel planning scenario, serving real online users, with an accuracy of 84\%. We believe this evidence can better demonstrate the effectiveness of ToolACE.
>
> **W2:** **The self-guided dialog generation process heavily relies on the LLM being trained to evaluate the complexity of generated data. This creates a circular dependency where the model is used both as a learner and an evaluator, which may introduce bias in the complexity estimation. More external validation or use of independent evaluators would make the results more robust.**
>
> **A2:** The self-guided method is based on the assumption that the most appropriate training data is contingent on the current capabilities of the trained model. Thus, we intentionally utilize the model both as a learner and an evaluator. While previous research has drawn similar conclusions [1,2], we acknowledge that additional validation is beneficial. Hence we conduct an additional experiment in our revised manuscript in Appendix E.3, where we use an independent model (Qwen1.5-7B-Chat, selected to maintain a comparable size for fairness) as the evaluator. The results, presented in the table below, indicate that using the model being trained as the complexity evaluator offers more accurate guidance, leading to improved performance on the BFCL benchmark.
>
> **Table 1. Ablation study on complexity evaluator.**
> | Evaluator | Learner | Overall | Non-live(A) | Non-live(E) | Live(A) | Multi turn | Rel | Irrel|
> | -------- | -------- | -------- | -------- | -------- | -------- | -------- | -------- | -------- |
> | Qwen1.5-7B-Chat | LLaMA-3.1-8B-Instruct |57.61 | 90.42 | 85.88 | 71.30 |13.12 | 87.80 | 78.12 |
> | LLaMA-3.1-8B-Instruct | LLaMA-3.1-8B-Instruct | 59.22 | 89.27 | 90.07 | 73.21 | 14.37 | 85.37 | 83.81 |
>
>
> [1] Du et al. 2023, MoDS: Model-oriented Data Selection for Instruction Tuning
> [2] Ren et al. 2024, Learning or Self-aligning? Rethinking Instruction Fine-tuning
>
> **W3:** **The use of complexity-based sampling to dynamically adjust dialog difficulty has merit but may lead to unintended biases, as data that is either too simple or too complex is filtered out. The approach may fail to fully explore the impact of diverse and extreme cases, leading to gaps in the model’s capabilities in certain contexts.**
>
> **A3:** As shown in Section 3.3.2, our results indicate that the model performs optimally when trained on a medium-complexity subset of data, suggesting that both overly simple and overly complex data are less effective for model training. While the model’s foundational knowledge likely stems from its pre-training stage, we believe that the key to improving model performance during fine-tuning is identifying the most appropriate training set. If the data is too simple for itself, the model has already mastered the ability and has no need to learn it again. If the data is too complex for the model, its prediction can still be incorrect even if similar data has been trained in the finetuning phase.
>
> We acknowledge that non-uniform sampling can introduce bias, such as causing the model to struggle with learning difficult examples after one round of training, effectively remaining in its "comfort zone." However, based on previous studies[3], iteratively training the model using samples generated by the model itself has been shown to extend the model's knowledge boundary, achieving effects akin to curriculum learning. In future work, we will further explore the proposed complexity-based sampling strategy to perform iterative training and sampling over multiple rounds, thereby progressively enhancing the model's generalization capability on more challenging samples.
>
> [3] Huang, Jiaxin, et al. "Large language models can self-improve." EMNLP(2023).

---

> ### Author Response · Authors · 2024-11-17
> **Responses to Weakness 4, 5, 6**
>
> **W4:** **While the paper compares ToolACE to several other function-calling models, the comparison is often superficial. The benefits of using ToolACE versus simpler data augmentation techniques are not well articulated, and it is unclear how much of the improvement can be attributed to the synthesis method versus the increased volume of data.**
>
> **A4:** Thanks for the valuable point. We recognize the importance of a more fair comparison and have conducted additional experiments under a controlled setting in revision in Appendix E.1. Specifically, we compare the performances of using ToolACE and other state-of-the-art function-calling data (ToolLLM and xLAM) to train the same base model (Llama3.1-8B). All data is uniformly sampled to 25,000 for a fair comparison. The corresponding results on BFCL are presented in the table below. These results demonstrate that the model trained with our data consistently outperforms the other models in all categories, further validating the effectiveness of our approach. Notably, the model trained on the xLAM dataset exhibits relatively poor performance in irrelevance detection, likely due to a lack of diverse sample types, such as cases where provided tools cannot solve the task. Moreover, the ToolLLM dataset, which primarily focuses on multi-step and dependent cases, demonstrates weak generalization on the BFCL benchmark.
>
> **Table 2. Performances of training with different training datasets.**
> | Training data | Overall | Non-live(A) | Non-live(E) | Live(A) | Multi turn | Rel | Irrel|
> | -------- | -------- | -------- | -------- | -------- | -------- | -------- | -------- |
> | ToolLLM(2.5w) | 24.90 |  42.46 | 36.36 | 39.45 | 0.00 | 100.00 | 4.41 |
> | xLAM(2.5w)    | 40.51 | 81.94 | 81.77 | 43.18 | 4.38 | 73.17 | 11.87 |
> | ToolACE(2.5w) (Ours) | **58.19** | **86.96** | **84.73** | **71.35** | **16.50** | **75.61** | **86.42** |
>
> **W5:** **The paper claims that ToolACE-8B is competitive with GPT-4 series models. However, it does not fully address the limitations of ToolACE-8B in terms of generalization and applicability to a broader range of tasks beyond function calling. A more detailed discussion of these limitations would provide a more balanced perspective.**
>
> **A5:** We thank the reviewer for the valuable suggestion. ToolACE-8B is specifically designed for function calling tasks. We highlight the competitiveness of ToolACE in the domain it was designed to excel at. Meanwhile, to show the generalization ability of ToolACE, we have conducted experiments to assess ToolACE-8B’s performance on broader tasks in Section 3.6. The results, shown in Figure 8, demonstrate that ToolACE-8B maintains competitive performance relative to its base model, Llama3.1-8B-Instruct, across general tasks such as coding, math, and reasoning. Our findings highlight that a smaller, specialized model like ToolACE-8B can outperform a more generalized model like GPT-4 in areas where it is specifically optimized, while still preserving robust general performance.
>
> Our experiments demonstrate that ToolACE-generated data can significantly enhance function-calling capabilities without substantially compromising other abilities. However, as you pointed out, we have not investigated how to simultaneously improve other capabilities alongside function-calling performance, which remains an open question in the field. This issue is beyond the scope of this paper, as our primary goal is to develop a specialized model for function calling. Nonetheless, our data synthesis approach may offer insights for other domains, such as strategies to enhance data accuracy, diversity, and complexity.
>
>
> **W6:** **The font size in Figures is too small, which is unclear for readers.**
>
> **A6:** Thanks for the suggestion! We have updated the figures in our revised manuscript (uploaded to the system).

---

> ### Comment · Reviewer_ViF9 · 2024-11-20
>
> Thank you for your response. However, it did not address my concern that "the self-guided dialog generation process heavily relies on the LLM being trained to evaluate the complexity of the generated data."
>
> "A more detailed discussion of these limitations would provide a more balanced perspective" remains unaddressed.
>
>  I would prefer to retain my scores.

---

> > ### Author Response · Authors · 2024-11-23
> > **Responses to the concerns**
> >
> > Thank you for your response. We would like to take this chance to make further clarifications about the two concerns. If there are any remaining unclear points, we would be more than happy to further clarify them during the discussion.
> >
> > **Concern about the Data Complexity Evaluator**
> >
> > To address your concerns regarding our data complexity evaluator, we have extended our analysis by conducting experiments using Qwen1.5-7B-Chat and Qwen1.5-14B-Chat as complexity evaluators. The results, summarized in Table 1 below, indicate that using the model being trained as the complexity evaluator provides more effective guidance, leading to improved performance on the BFCL benchmark.
> >
> > Notably, when the complexity scores are assessed using a more advanced model (e.g., Qwen-14B), certain simpler training samples—those marked as "easy" by the evaluator but not necessarily by the learner—may be excluded. This exclusion leads to slight performance gains on more challenging tasks (e.g., Live AST) but results in performance degradation on Non-live AST tasks$^1$. Conversely, when the evaluator is less capable than the learner, the retained samples tend to be relatively easier for the learner, leading to better results on Non-live AST tasks while causing a decline in performance on Live AST tasks.
> >
> > We acknowledge that our current self-guided evaluator may not be the ideal solution for identifying the most suitable training set. For instance, it may be sensitive to model size, and scaling it up for larger datasets may present challenges. We have also discussed the potential bias introduced in W3 in our previous responses. However, the experiments we conducted demonstrate that the current approach is both effective and reasonably appropriate. If these concerns have not yet been fully addressed, we would appreciate further clarification on the specific aspects you believe require more attention.
> >
> > **Table 1. Ablation study on complexity evaluator.**
> > | Evaluator | Learner | Overall | Non-live(A) | Non-live(E) | Live(A) | Multi turn | Rel | Irrel|
> > | -------- | -------- | -------- | -------- | -------- | -------- | -------- | -------- | -------- |
> > | Qwen1.5-7B-Chat | LLaMA-3.1-8B-Instruct |57.61 | 90.42 | 85.88 | 71.30 |13.12 | 87.80 | 78.12 |
> > | Qwen1.5-14B-Chat | LLaMA-3.1-8B-Instruct |57.67 | 87.98 | 87.02 | 73.30 |11.75 | 87.80 | 84.00 |
> > | LLaMA-3.1-8B-Instruct | LLaMA-3.1-8B-Instruct | 59.22 | 89.27 | 90.07 | 73.21 | 14.37 | 85.37 | 83.81 |
> >
> >
> > > $^1$ Live AST tasks involve rarer and more complex functions compared to Non-live AST tasks, as detailed in BFCL's documentation.
> >
> > **Concern about Analysis of Limitations**
> >
> > While we have conducted extensive experiments to demonstrate the effectiveness of our synthesized dataset in enhancing functional-calling performance, several challenges remain.
> >
> > - Computational Complexity: The data complexity evaluation is influenced by the size of the model being trained, which limits scalability as both the model size and the number of training samples increase. Despite this, we believe the approach we have evaluated remains an effective strategy.
> >
> > - Model Performance Limitations: Although our model shows strong performance in functional calling, it still lags behind GPT-4 in other capabilities. To compare general capabilities, we have evaluated GPT-4 across several benchmarks, with results presented in Figure 8 of the revised manuscript. As expected, our ToolACE-8B model performs below GPT-4 in areas such as reasoning, mathematics, and coding. This is primarily due to the scale of the model and its training corpus. When compared to its base model, LLaMA-3.1-8B-Instruct, ToolACE-8B demonstrates substantial improvements in functional calling with minimal negative impact on other capabilities. While this success highlights the potential of specialized models in one specific domain, the challenge of simultaneously enhancing multiple capabilities, alongside functional-calling performance, remains an open question.
> >
> > The analysis about the limitations are all updated in our revised manuscript in Appendix H.

---

> > > ### Author Response · Authors · 2024-11-25
> > > **Follow-Up on Second-Round Responses to Your Comments**
> > >
> > > Dear Reviewer ViF9,
> > >
> > > As the end of the discussion period is approaching, we would like to kindly follow up to see if you have had a chance to review our second-round responses to your comments. We sincerely appreciate the valuable feedback you have provided, which has been essential in improving our work and providing a more balanced perspective. We have carefully addressed your suggestions and would be happy to clarify or address any remaining concerns you might have.

---

> > > > ### Comment · Reviewer_ViF9 · 2024-11-25
> > > >
> > > > Thank you for your responses. I have updated my score.

---

### Official Review · Reviewer_AcuD · 2024-11-03

**Soundness:** 3
**Presentation:** 2
**Contribution:** 2
**Rating:** 6
**Confidence:** 4

**Summary:**

Great idea and well written. The authors address the challenges of collecting accurate, diverse, and complex tool-usage data for LLMs by introducing a novel self-evolution synthesis process. ToolACE synthesizes a comprehensive API pool and generates data through agent-based dialogues, guided by a complexity evaluator to ensure the difficulty level is suited to the model's capabilities. The paper presents dual-layer verification (DLV) to maintain data quality, combining rule-based checks and model-based validation. Experiments demonstrate that models trained on ToolACE data achieve state-of-the-art performance in function calling, outperforming models such as GPT-4 in specific benchmarks like BFCL and APIBank.

**Strengths:**

ToolACE introduces a unique self-evolution synthesis method, which is a systematic approach to generating diverse and complex data for function calling, addressing a key limitation in existing tool-augmented LLMs. The paper provides extensive experiments and ablation studies, comparing ToolACE-trained models with existing benchmarks on widely used datasets like BFCL and APIBank, and demonstrating superior performance.

**Weaknesses:**

1. Please include a complete example of a prompt and LLM response in the appendix so that readers can intuitively understand how the process works in practice.

2. The paper lacks clarity and involves overly complex technical concepts. Although constructing a simulated dataset and fine-tuning the model are effective approaches to enhancing the LLM's function call capabilities, the additional concepts introduced, such as Self-Evolution, Self-Guided, Dual-Layer, and Multi-Agent, make the main idea harder to discern, leading to confusion for the reader. While the authors may believe these terms add richness to the paper, they detract from its central focus.

3. In the ablation study, it would be valuable to compare the Retrieval-Augmented Generation (RAG) approach for retrieving task-relevant tools with In-Context Learning to optimize tool usage. Given the same level of engineering effort, explore whether these methods could achieve results comparable to fine-tuning.

**Questions:**

In Figure 3, the "without model" approach occasionally outperforms the "with model" approach. Please provide an analysis to explain the reasons for this phenomenon.

---

> ### Author Response · Authors · 2024-11-17
> **Responses to Weaknesses and Questions**
>
> **W1:** **Please include a complete example of a prompt and LLM response in the appendix so that readers can intuitively understand how the process works in practice.**
>
> **A1:** We thank the reviewer for the helpful suggestion to make our manuscript clearer. A revision has been updated in the system with a complete example of a prompt and an LLM response, as shown in Appendix F. We also include a detailed explanation of the two benchmarks in Appendix C.1.
>
> **W2:** **The paper lacks clarity and involves overly complex technical concepts. The additional concepts introduced, such as Self-Evolution, Self-Guided, Dual-Layer, and Multi-Agent, make the main idea harder to discern, leading to confusion for the reader. While the authors may believe these terms add richness to the paper, they detract from its central focus.**
>
> **A2:** Thank you for your valuable feedback. We understand your concern about the complexity and the introduction of new concepts in the paper. Our central focus, as reflected in the title, is to improve the accuracy, complexity, and diversity of synthetic data to enhance the function-calling capabilities of LLMs. To achieve this, we propose specific methods for each dimension:
>
> - **Accuracy:** We introduce Dual-Layer data verification, which improves synthetic data accuracy by using both rule-based and model-based checkers.
> - **Complexity:** We propose Self-Guided dialog generation, where a newly defined complexity evaluation metric helps guide the dialog generation process.
> - **Diversity:** We present Tool Self-Evolution Synthesis, a method that generates a large scale of diverse tools through a self-evolution process to increase data diversity.
>
> Regarding Multi-Agent dialog generation, this is part of our Self-Guided approach, and while it is a widely used technique for generating synthetic dialogs, it is not one of our primary contributions.
>
> We hope this clarifies the structure of our approach and how each method contributes to the overall goal.
>
> **W3:** **In the ablation study, it would be valuable to compare the Retrieval-Augmented Generation (RAG) approach for retrieving task-relevant tools with In-Context Learning to optimize tool usage. Given the same level of engineering effort, explore whether these methods could achieve results comparable to fine-tuning.**
>
> **A3:** In this paper, we focus primarily on the **tool calling** capability of LLMs, where candidate tools are already provided as part of the model's input. While enhancing tool retrieval through RAG methods could be valuable in real-world applications, it is outside the scope of this study, as it addresses a different aspect of the overall tool-usage pipeline (i.e., tool retrieval vs. tool calling). Regarding the comparison between in-context learning and fine-tuning for optimizing tool usage, we acknowledge that in-context learning can be a viable alternative, but its effectiveness is highly dependent on the model’s initial capabilities. We have added experimental comparison and discussion on it in Appendix G of our revised manuscript. Specifically, we use the training samples as few-shot candidates and retrieve the top 3 most relevant samples according to the user's question and the provided tools with the BGE model to guide in-context learning. The results below show that few-shot in-context learning not only underperforms fine-tuning in BFCL but also falls short of the zero-shot setting. In many cases, the model is misled by the tools in the few-shot examples, selecting those instead of the tools in the test sample, which further exacerbates the model's hallucination phenomenon, such as the example illustrated in Figure 16 in Appendix G.
>
> **Table 1. Comparison between in-context learning and finetuning**
> | Model | Non-live(A) | Non-live(E) | Live(A) | Rel | Irrel|
> | -------- | -------- | -------- | -------- | -------- | -------- |
> | Llama-3.1-8B-Instruct (Zero-shot) | 86.23 | 83.48 | 48.02 | 82.93 | 23.05 |
> | Llama-3.1-8B-Instruct (3-shot) | 58.81 | 53.32 | 36.83 | 82.93 | 23.66 |
> | ToolACE-8B (Ours) | 89.27 | 90.07 | 73.21 |  85.37 | 83.81 |
>
>
> **Q1:** **In Figure 3, the "without model" approach occasionally outperforms the "with model" approach. Please provide an analysis to explain the reasons for this phenomenon.**
>
> **A4:** The model verification layer depends on the model's ability to identify errors in the data. As a result, false negatives—where data without any errors is incorrectly flagged as erroneous—can occur. These false negatives may cause occasional performance drops in some categories. Despite this, the performance decline in these cases is generally not significant, and we consider it to be within an acceptable tolerance. The overall improvement provided by the model verification layer still outweighs these occasional discrepancies.

---

> > ### Comment · Reviewer_AcuD · 2024-11-19
> > **Thank you for your feedback**
> >
> > For W1 and W3, the response is good.
> >
> > For W2 and A2, Maybe you think that introducing a lot of concepts (e.g., fancy adjectives) will make the paper more innovative, but from the reader's point of view, it will only distract from what the paper is trying to argue. No matter how much you try to explain.
> >
> > I will keep my score and hope I good results.
> >
> > Thanks.

---

> > > ### Author Response · Authors · 2024-11-19
> > > **Thank you**
> > >
> > > We thank the reviewer for the positive score. We're glad to learn that most of the concerns have been solved, and we'll make sure the writing is improved in our final version!

---

### Official Review · Reviewer_Pz3q · 2024-11-07

**Soundness:** 2
**Presentation:** 2
**Contribution:** 2
**Rating:** 6
**Confidence:** 4

**Summary:**

This paper tries to improve the function calling capability of LLM by finetuning on a newly collected function calling dataset. Specifically, the authors propose a pipeline to collect new API usage data. This dataset is then used to finetune llama 8b model and shows comparative performances on two API using benchmarks.

**Strengths:**

- Collecting new data in a scalable way is important
- The performance looks interesting as well by finetuning a small model

**Weaknesses:**

The experiment section is not quite convincing yet. Since the authors want to show the effectiveness of using their newly collected API (which according to table 1) is much more comprehensive, the authors should compare the performance obtained by finetuning on Table 1 datasets e.g. ToolLLM and that obtained by finetuning on their newly collected API data

**Questions:**

- As mentioned in weakness, additional experiment with other baselines should be included e.g. ToolLLM. Even the authors provide some results of xLAM in Table 2, I noticed that they are tuned on different base models other than 8b. So it is hard to draw conclusions and give credit to the dataset itself or to the base models. According to Table 3, 8b seems already very good at API calling evaluations.
- The evaluation and experiment process is not quite clear. e.g. what are the benchmark APIBank, BFCL evaluating? What are their input, output, ground truth, etc? What is the metric used in Table 2,  Table 3?
- Table 2 has many categories in the performances: Single Turn, Multi Turn, Live, Hallucination. Compared to the base model used by author (llama 8b), some categories have only limited performance gain while some could be much higher due to finetuning (Multi turn), thus leading to a higher average score. I don't understand these comparison categories and I don't see authors' analysis in that.

---

> ### Author Response · Authors · 2024-11-17
> **Response to Question 1**
>
> **Q1:** **Performances obtained by finetuning the same base model on Table 1 datasets e.g. ToolLLM.**
>
> **A1:** Thank you for your valuable suggestion. We did not include results for using the datasets in Table 1 to train the same model primarily because most related works optimize their fine-tuned models by choosing base models, data, and training settings. However, we recognize that including these results can further demonstrate the value of our generated data, which is a good complement to the existing results. Hence we have conducted additional experiments under a controlled setting in our revised version in Appendix E.1. Specifically, we compare the performances of using ToolACE and other state-of-the-art function calling data mentioned in Table 1 (ToolLLM and xLAM) to train the same base model (Llama3.1-8B). All data are uniformly sampled to 25,000 for a fair comparison. The corresponding results on BFCL are presented in the table below. These results demonstrate that the model trained with our data consistently outperforms the other models in all categories, further validating the effectiveness of our approach. Notably, the model trained on the xLAM dataset exhibits relatively poor performance in irrelevance detection, likely due to a lack of diverse sample types, such as cases where provided tools cannot solve the task. Moreover, the ToolLLM dataset, which primarily focuses on multi-step and dependent cases, demonstrates weak generalization on the BFCL benchmark.
>
>
> **Table 1. Performances of training with different training datasets.**
> | Training data | Overall | Non-live(A) | Non-live(E) | Live(A) | Multi turn | Rel | Irrel|
> | -------- | -------- | -------- | -------- | -------- | -------- | -------- | -------- |
> | ToolLLM(2.5w) | 24.90 |  42.46 | 36.36 | 39.45 | 0.00 | 100.00 | 4.41 |
> | xLAM(2.5w)    | 40.51 | 81.94 | 81.77 | 43.18 | 4.38 | 73.17 | 11.87 |
> | ToolACE(2.5w) (Ours) | **58.19** | **86.96** | **84.73** | **71.35** | **16.50** | **75.61** | **86.42** |

---

> ### Author Response · Authors · 2024-11-17
> **Response to Question 2**
>
> **Q2:** **What are the benchmark APIBank, BFCL evaluating? What are their input, output, ground truth, etc? What is the metric used in Table 2, Table 3?**
>
> **A2:** Both the BFCL and API-Bank benchmarks assess the function-calling capabilities of LLMs, but they differ slightly in their evaluation setups.
>
> - **BFCL**
>   The Berkeley Function-Calling Benchmark (BFCL) is a comprehensive evaluation framework for assessing the function-calling capabilities of LLMs across various languages, application domains, and complex use cases. BFCL covers tasks including multiple function calls, parallel function calls, multi-turn function calls, and multi-step function calls. BFCL contains 4,951 test cases: 3,951 single-turn cases and 1,000 multi-turn cases, focusing on dynamic, real-world scenarios.
>
>   BFCL splits the evaluation into three categories:
>   - **Single-turn:** Evaluating function calls in a single interaction. Single-turn is further evaluated in three settings: non-live (AST), non-live (Executable), and live (AST). Non-live (AST) compares the abstract syntax tree of the function output to the ground truth and the function definition. Non-live (Executable) assesses the accuracy of the generated API call by executing it and comparing the output with the ground-truth output. Live (AST) employs live, user-contributed function documentation and queries with abstract syntax tree comparison, avoiding the drawbacks of dataset contamination and biased benchmarks.
>   - **Multi-turn:** Evaluating the ability to maintain state and make function calls across multiple interactions, making it possible for LLMs to navigate through complex tasks by asking clarifying questions.
>   - **Hallucination:** Evaluating whether the model generates irrelevant or incorrect responses, rather than valid function calls. Hallucination is further categorized into relevance and irrelevance detection. Relevance evaluates the model's ability to output function calls relevant to the user query. Irrelevance measures the model's ability to refrain from making function calls given irrelevant user queries.
>
> - **API-Bank**
>   API-Bank consists of 314 tool-use dialogues with 753 API calls to assess LLMs’ capabilities in planning, retrieving, and calling APIs, with 363 single calls and 122 multiple calls. API-Bank assesses LLM performance across two capabilities:
>
>   - **Call:** The ability to call an API based on a given query when the APIs are known.
>   - **Retrieval+Call:** The ability to retrieve and call a single API when the APIs are unknown.
>
>
> **Common Input, Output, and Ground Truth for Both Datasets:**
> - **Input:** The model receives a list of candidate tools (e.g., available APIs or functions) and the conversation history, which includes the user’s request or query.
> - **Output:** The expected output is the model’s function call (e.g., an API request) or, in the case of hallucinations, an irrelevant or erroneous response in natural language.
> - **Ground Truth:** The ground truth is the correct function call, which is either determined by matching the generated function call with a predefined correct one or by checking the execution results for valid function calls (if execution is possible).
>
>
> **Evaluation Metric:**
> The metric used in Table 2 and Table 3 is **accuracy**, which measures the proportion of correct function calls generated by the model, as compared to the ground truth.
>
> A more precise explanation of the benchmarks is included in **Appendix C.1** in our revision.

---

> > ### Author Response · Authors · 2024-11-17
> > **Examples of BFCL and API-Bank**
> >
> > ### BFCL example
> > ```
> > System: You are an expert in composing functions. You are given a question and a set of possible functions.
> > Based on the question, you will need to make one or more function/tool calls to achieve the purpose. If none of the functions can be used, point it out. If the given question lacks the parameters required by the function, also point it out. You should only return the function call in the tools call sections.
> > If you decide to invoke any of the function(s), you MUST put it in the format of [func_name1(params_name1=params_value1, params_name2=params_value2...), func_name2(params)]
> > You SHOULD NOT include any other text in the response.
> >
> > Here is a list of functions in JSON format that you can invoke:
> >
> > [{"name": "get_weather_data", "description": "Fetches weather data from the Open-Meteo API for the given latitude and longitude.", "parameters": {"type": "dict", "properties": {"coordinates": {"type": "array", "items": {"type": "float"}, "description": "The latitude and longitude of the location."}}, "required": ["coordinates"]}}, {"name": "calc_binomial_probability", "description": "Calculates the probability of getting k successes in n trials.", "parameters": {"type": "dict", "properties": {"n": {"type": "integer", "description": "The number of trials."}, "k": {"type": "float", "description": "The number of successes."}, "p": {"type": "float", "description": "The probability of success."}}, "required": ["n", "k", "p"]}}]
> >
> >
> > User: I'm planning a small outdoor event in Ottawa, and I need to make sure the weather is going to cooperate. Could you fetch the current weather for me at latitude 45.4215 and longitude -75.6972 using the Open-Meteo API? Also, I'm running a small game at the event, and I'm curious about the chances of winning. If I have 10 attempts at this game and the chance of winning each time is 50\%, how likely is it that I'll win 5 times?
> >
> > Assistant (expected output): [get_weather_data(coordinates=[45.4215, -75.6972]), calc_binomial_probability(n=10, k=5.0, p=0.5)]
> > ```
> >
> >
> > ### API-Bank example
> > ```
> > System: Based on the given API description and the existing conversation history 1..t, please generate the API request that the AI should call in step t+1 and output it in the format of [ApiName(key1='value1', key2='value2', ...)], replace the ApiName with the actual API name, and replace the key and value with the actual parameters.
> > Your output should start with a square bracket "[" and end with a square bracket "]". Do not output any other explanation or prompt or the result of the API call in your output.
> > This year is 2023.
> >
> > Input:
> > User: [User's plain text or response of API call]
> > AI: [AI's plain text]
> > ...
> > User: [User's plain text or response of API call]
> >
> > Expected output:
> > [ApiName(key1='value1', key2='value2', ...)]
> >
> > API descriptions:
> > [{"name": "GetUserToken", "description": "Get the user token by username and password.", "input_parameters": {"username": {"type": "str", "description": "The username of the user."}, "password": {"type": "str", "description": "The password of the user."}}, "output_parameters": {"token": {"type": "str", "description": "The token of the user."}}}, {"name": "AddAlarm", "description": "The API for setting an alarm includes a parameter for the alarm time.", "input_parameters": {"token": {"type": "str", "description": "User"s token."}, "time": {"type": "str", "description": "The time for alarm. Format: %Y-%m-%d %H:%M:%S"}}, "output_parameters": {"status": {"type": "str", "description": "success or failed"}}} ]
> >
> >
> > User: Can you help me set an alarm for 8 am tomorrow?
> >
> > Assistant: Sure, to set an alarm, I need to authorize your account. Can you please provide your username and password?
> >
> > User: Sure, my username is foo and password is bar.
> >
> > Assistant (expected output): [GetUserToken(username="foo", passward="bar")]
> >
> > Tool: [GetUserToken Response: {"token": "z9x8c7v6b5n4m3q2w1"}]
> >
> > Assistant: Okay, I got your token. What time do you want the alarm to be set for?
> >
> > User: 8 am tomorrow. Today is 2021-10-13.
> >
> > Assistant (expected output): [AddAlarm(token="z9x8c7v6b5n4m3q2w1", time="2021-10-14 08:00:00")]
> >
> > Tool: [AddAlarm Response: ”success"]
> >
> > Assistant: An alarm has been set for 8 am tomorrow.
> > ```

---

> ### Author Response · Authors · 2024-11-17
> **Response to Question 3**
>
> **Q3:** **Compared to the base model used by author (Llama 8B), some categories have only limited performance gain while some could be much higher due to fine-tuning (Multi-turn), thus leading to a higher average score. I don't understand these comparison categories.**
>
> **A3:** Compared with the Llama-3.1-8B-Instruct, the model after fine-tuning achieves significant improvements over all categories, and the improvement on multi-turn is much higher, as illustrated in Table 2. This can be attributed to two major reasons:
> First, the multiple rounds function calling is more challenging than single-turn samples, and the probability of failure will be exponential of the single-turn calling (i.e. the error rate will increase from $p$ (single-turn) to $1-(1-p)^n$. Second, benefiting from the diversity of the ToolACE dataset, multi-turn, and dependent samples are included in the training data, enhancing the model's ability in dealing with such long-context problems.
>
> Additionally, the gain of irrelevance detection (Irrel) is also high, this may be because Irrel measures the model's ability to NOT call functions given irrelevant user queries, which requires the model to precisely understand the user intents and the tool functionality. The Irrel score reflects a different aspect of function calling—it measures the model's ability to withhold action when inappropriate, which requires a higher level of judgment and comprehension. While ToolACE covers a more diverse set of examples, including multi-type data with both function calling data and non-tool data, thus better helping the model in learning when to refrain from invoking a tool.
>
> **Table 2. Performance improvements on various categories.**
> | Model | Overall | Non-live(A) | Non-live(E) | Live(A) | Multi turn | Rel | Irrel|
> | -------- | -------- | -------- | -------- | -------- | -------- | -------- | -------- |
> | Llama-3.1-8B-Instruct | 43.80 | 86.23 | 83.48 | 48.02 | 05.12 | 82.93 | 23.05 |
> | ToolACE-8B (Ours) | 59.22 | 89.27 | 90.07 | 73.21 | 14.37 | 85.37 | 83.81 |

---

### Meta-Review · Area_Chair_fjTy · 2024-12-18

**Metareview:**

This paper proposes a method to improve tool-calling for LLMs by generating a dataset using a multi-agent framework. The paper mainly focuses on this data creation part, which consists of three components (Tool Self-evolution Synthesis, Self-Guided Dialog Generation, and Dual-Layer Validation Process) -- which uses LLM agents and evaluators. The goal is to create accurate, complex, and diverse data for training LLMs to perform function-calling. Experiments on BFCL and API-Bank demonstrate that ToolACE-8B (trained using supervised fine-tuning from LLaMA3.1-8B-Instruct) achieves promising results, even compared to GPT-4, on these specific benchmarks. The reviewers and AC appreciate the importance of curating new data as well as the promising results from the LLM trained using this dataset. The weakness of the paper is that the pipeline to create the dataset is complex with many details and reviewers raised concerns on the depth of the experiments (e.g. how does performance scale with data). Ultimately, despite the weaknesses, there is unanimous decision to accept this work, and I think the demonstration of the value of data for function-calling will have broad interest and significance to the wider community.

**Additional Comments On Reviewer Discussion:**

Reviewers initially raised concerns on the experimental design, complexity of the data generation pipeline, and clarifications (e.g. on the prompt). The authors were able to provide detailed responses. A few questions from reviewers remain after and the AC highly encourages the authors to take this into account even after acceptance (e.g. complexity of the pipeline itself, json/code representation, additional experiments to better understand impact of the amount/type training data). All reviewers had unanimous decision to accept after the rebuttal period.

---

### Decision · Program_Chairs · 2025-01-22

Accept (Poster)